# Rat behavior and dopamine release are modulated by conspecific distress

Nina T Lichtenberg[1†], Brian Lee[1†], Vadim Kashtelyan[1], Bharadwaja S Chappa[2], Henok T Girma[2], Elizabeth A Green[2], Shir Kantor[2], Dave A Lagowala[2], Matthew A Myers[2], Danielle Potemri[2], Meredith G Pecukonis[2], Robel T Tesfay[2], Michael S Walters[2], Adam C Zhao[1], R James R Blair[3], Joseph F Cheer[4,5,6], Matthew R Roesch[1,2,7]*

[1]Department of Psychology, University of Maryland, College Park, United States; [2]Gemstone Honors Program, University of Maryland, College Park, United States; [3]Center for Neurobehavioral Research, Boys Town National Research Hospital, Boys Town, United States; [4]Department of Anatomy and Neurobiology, University of Maryland School of Medicine, Baltimore, United States; [5]Department of Psychiatry, University of Maryland School of Medicine, Baltimore, United States; [6]Program in Neuroscience, University of Maryland School of Medicine, Baltimore, United States; [7]Program in Neuroscience and Cognitive Science, University of Maryland, College Park, United States

*For correspondence:
mroesch@umd.edu

[†]These authors contributed equally to this work

Competing interests: The authors declare that no competing interests exist.

**Abstract** Rats exhibit 'empathy' making them a model to understand the neural underpinnings of such behavior. We show data consistent with these findings, but also that behavior and dopamine (DA) release reflects subjective rather than objective evaluation of appetitive and aversive events that occur to another. We recorded DA release in two paradigms: one that involved cues predictive of unavoidable shock to the conspecific and another that allowed the rat to refrain from reward when there were harmful consequences to the conspecific. Behavior and DA reflected pro-social interactions in that DA suppression was reduced during cues that predicted shock in the presence of the conspecific and that DA release observed on self-avoidance trials was present when the conspecific was spared. However, DA also increased when the conspecific was shocked instead of the recording rat and DA release during conspecific avoidance trials was lower than when the rat avoided shock for itself.
DOI: https://doi.org/10.7554/eLife.38090.001

## Introduction

The ability to appropriately adjust behavior in a social context is performed daily and is impacted in several psychiatric disorders (*Baron-Cohen et al., 1985*; *Blair, 2003*; *Cook and Black, 2012*; *Gaigg, 2012*; *Hamilton, 2013*; *Matthys et al., 2013*; *Neuhaus et al., 2010*; *Rizzolatti and Fabbri-Destro, 2008*; *Scott-Van Zeeland et al., 2010*; *Taylor and DeQuinzio, 2012*). However, it is unclear how neurobiological substrates integrate harm of others into our decisions because detailed work in animals at the single-unit and neurotransmitter level has not yet occurred. A growing number of high-profile reports have demonstrated that rodent models can be used to perform such experiments (*Atsak et al., 2011*; *Ben-Ami Bartal et al., 2011*; *Ben-Ami Bartal et al., 2014*; *Burkett et al., 2016*; *Church, 1959*; *Guzmán et al., 2009*; *Hernandez-Lallement et al., 2018*; *Hernandez-Lallement et al., 2018*; *Hernandez-Lallement et al., 2017*; *Jeon et al., 2010*; *Jones et al., 2014*; *Kashtelyan et al., 2014*; *Kim et al., 2010*; *Langford et al., 2006*; *Masserman et al., 1964*; *Meyza et al., 2017*; *Mogil, 2015*; *Panksepp, 2011*; *Panksepp and Lahvis, 2011*; *Sato et al., 2015*;

*Schaich Borg et al., 2017*). For example, rats have been described as 'empathetic' and 'prosocial' because they freeze when they see others freeze in response to threat and will rescue conspecifics from distress when given the opportunity (*Atsak et al., 2011*; *Ben-Ami Bartal et al., 2011*; *Ben-Ami Bartal et al., 2014*; *Meyza et al., 2017*). These prosocial behaviors are particularly strong when rats are cage mates and in situations in which there are no explicit negative consequences to the rat performing the prosocial act.

Although we often perform prosocial actions that benefit others and suppress actions that might cause harm to others, there are other circumstances where this does not occur. This is especially true when our own sacrifice is too great or when the benefactor is not a friend, family member, or somebody believed to not reciprocate. How we respond to the distress of others is also modified by potential consequences for oneself. That is, we might feel distress when aversive events happen to somebody else, but in other contexts in which we are also at risk, we might be relieved that our own self-interests were not affected. Here, we show that rats can be more interested in their own physical state (subjective value) as opposed to another's (objective value), which we will refer to as 'self-interested'. Further, we demonstrate that subsecond dopamine (DA) release in nucleus accumbens core (NAc) accompanies this regard for one's own interest, reflecting the subjective value placed on appetitive and aversive events as opposed to the objective value of the event itself.

The DA system is unique in that it is the only neurotransmitter system that genuinely signals all aspects of reward prediction error encoding (*Bayer and Glimcher, 2005*; *Bromberg-Martin et al., 2010*; *Day et al., 2007*; *Gan et al., 2010*; *Hart et al., 2015*; *Hart et al., 2015*; *Roesch et al., 2010*; *Schultz, 2010*; *Schultz et al., 2015*; *Schultz et al., 1997*). By using a variety of technical approaches across species, studies have shown that DA neurons in the ventral tegmental area (VTA) and subsequent DA release into the NAc increases and decreases in response to events that are better or worse than expected, respectively. In the appetitive domain, cues that predict reward or the uncertain delivery of reward increase DA release, and cues that predict less favorable reward and omission of expected reward decrease DA release. In aversive contexts, cues predictive of unavoidable aversive events (e.g., quinine, air puff, shock) or delivery of those aversive outcomes themselves robustly reduce DA release, whereas unexpected omission of aversive events or the cues that predict avoidable shock reliably elicit phasic DA release (*McCutcheon et al., 2012*; *Oleson et al., 2012*; *Roitman et al., 2008*; *Wenzel et al., 2018*).

We therefore asked how DA signals are modulated by appetitive and aversive events that occur not to the animal from which recordings were obtained, but from a conspecific located nearby. To achieve this, we recorded accumbal subsecond DA release using fast-scan cyclic voltammetry (FSCV) in two different social paradigms: one that used Pavlovian cues to predict unavoidable shock to the recording rat or to a nearby conspecific; and another that had an instrumental component (lever press), which allowed the recording rat to refrain from the pursuit of reward when there were known harmful consequences (i.e., footshock) for itself or the conspecific.

In both studies, we found signs that rats were 'empathetic' and/or 'pro-social' as described above. Rats refrained from pressing for reward to spare the conspecific from shock and DA release mirrored trials when the rat we recorded from made the same decision for itself. Further, we show that when the recording rat was in the presence of the conspecific it displayed less fear and reduced DA inhibition to cues that predicted unavoidable shock. However, our results also show that rats are often not overly empathetic or prosocial; their behavior and DA signals were modulated by potential reward and shock that occurred directly to them as opposed to the conspecific, reflecting subjective rather than objective value. Specifically, we found that DA was released when the conspecific was shocked instead of the rat being recorded from and that DA signals reflecting the value of shock avoidance were weaker when the conspecific was spared.

## Results

### Pavlovian social distress paradigm

DA release was recorded from eight rats during performance of the Pavlovian Social Distress Paradigm, during which pairs of rats were placed side by side in a modified shuttle box, separated by a modified guillotine door that allowed rats to see, smell, and hear each other (*Figure 1A–D*). The exterior wall of each side was equipped with a directional cue light, a food cup, and a shock grid.

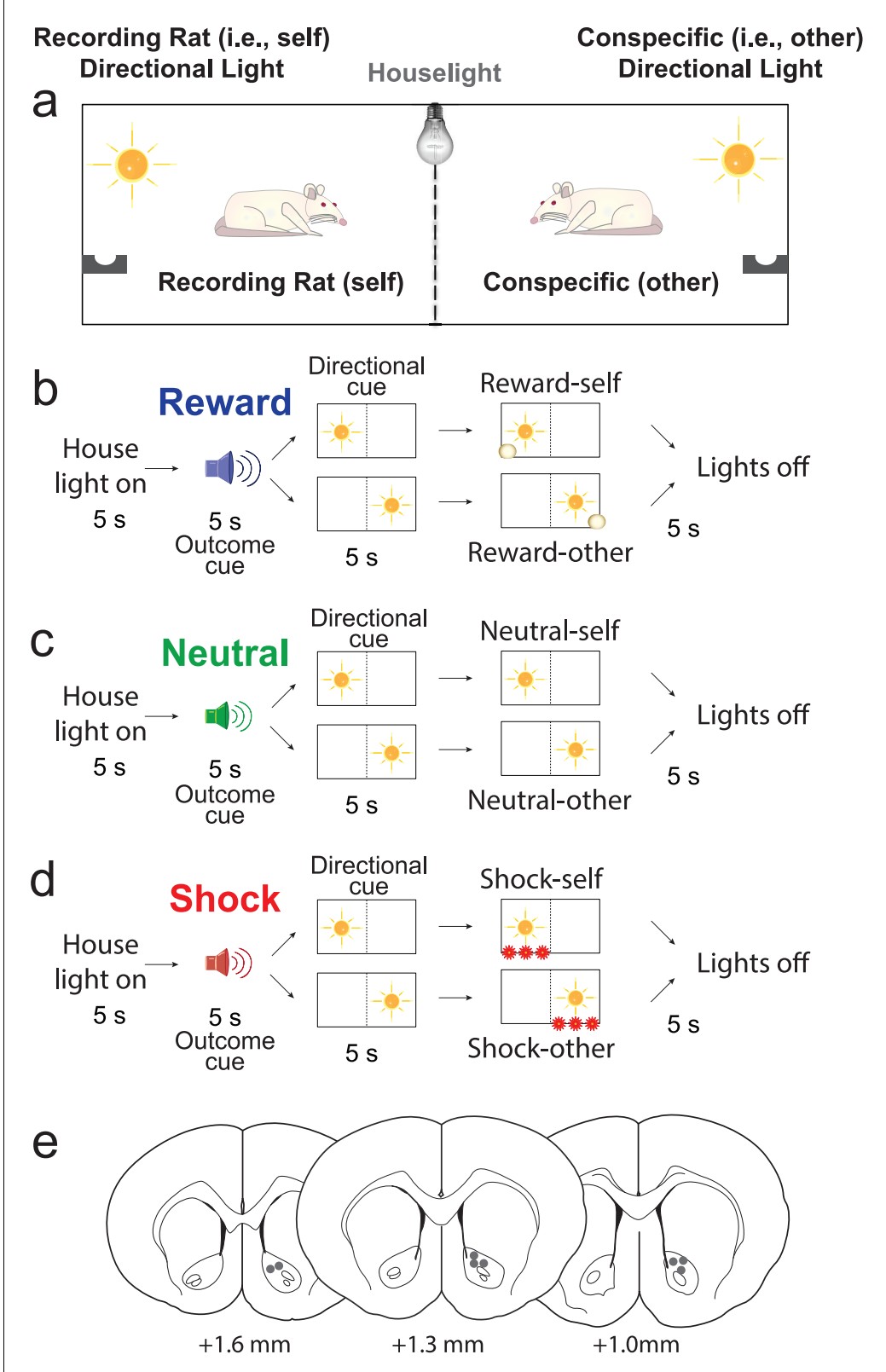

**Figure 1.** Pavlovian Social Distress Paradigm. (**A**) Rats were placed side by side, separated by a divider, during 'together' trial blocks. The conspecific was removed during 'alone' trial blocks. (**B–D**) Trial types. All trials started with illumination of a houselight. Five seconds later an auditory cue was presented indicating the outcome that would occur at the end of the trial (i.e., outcome cue). Five seconds after the 'outcome cue', the 'directional cue'

*Figure 1 continued*

was presented for 5 s, indicating which rat would receive the reward (i.e., reward trials; blue; **B**), nothing (i.e., neutral trials; green; **C**), or a shock (i.e., shock trials; red; **D**). (**E**) Placement of chronic recording electrodes within the NAc core based on histology.

DOI: https://doi.org/10.7554/eLife.38090.002

Each trial began with illumination of a house light. Five seconds after onset of the house light, three different auditory stimuli (5 s) predicted delivery of 3 different outcomes (sucrose pellet, footshock, or nothing) that could be delivered to either the recording rat or the conspecific. The odds were 50/50. Thus, during presentation of the auditory stimulus, animals did not know which rat would receive the outcome. This information was made known by subsequent illumination of one of the two directional light cues (5 s) located in either the recording rat's or the conspecific's chamber (*Figure 1A–D*). After presentation of the directional light for 5 s, the outcome (either reward, shock or nothing) was delivered. During each FSCV recording session (in NAc; *Figure 1E*) there were two trial blocks, one where both the conspecific and the recording rat were present and the other where the recording rat was alone (i.e., the conspecific's side of the box was empty).

When reporting the results pertaining to the Pavlovian Social Distress Paradigm we will adhere to the following terminology: 'Self' trials refer to trials during which the outcome (either reward, footshock, or neutral no outcome) was delivered to the recording rat, whereas 'other' trials refer to trials in which the outcome was delivered to the conspecific. 'Together' trials are defined as trials in which both the recording rat and conspecific were present, whereas 'alone' trials refer to trials in which only the recording rat was present. On 'alone' trials, the outcome was still delivered to the conspecific's side of the cage, even though the conspecific was absent. In the following figures, blue, green and red represent 'reward', 'neutral' and 'shock' trials, respectively, and the thickness of the line indicates whether the rat was alone (thin lines) or in the presence of the conspecific (thick lines). There were 12 trial-types in total (alone-reward-self, alone-reward-other, alone-neutral-self, alone-neutral-other, alone-shock-self, alone-shock-other, together-reward-self, together-reward-other, together-neutral-self, together-neutral-other, together-shock-self, together-shock-other).

## Rats correctly internalize auditory and directional light cues

Because the task was Pavlovian (i.e., no instrumental component) we used bream breaks in the food cup and video scoring to determine if recording rats understood the different trial-types. *Figure 2A and B* shows the average beam breaks into the food cup across all recording sessions (n = 40) for the three trial-types. Prior to outcome delivery, percent beam breaks respectively increased and decreased on reward-self (blue) and shock-self (red) trials relative to neutral (green) trials (*Figure 2A and B*). After the presentation of the directional light cue (i.e., the cue that informed the rat which animal would receive the outcome), there was a significant increase in beam breaks in the food cup for reward-self compared to reward-other trials (*Figure 2A* versus B; blue; Wilcoxon; $p < 0.05$). During shock trials, there was a general suppression across both 'self' (*Figure 2A*; red) and 'other' trial-types (*Figure 2B*; red), with little difference between them.

These effects are further illustrated and quantified in *Figure 2C–J* which depict distributions of the session difference beam break scores between shock and neutral trials (shock index = shock minus neutral) and between reward and neutral trials (reward index = reward minus neutral) for 'self' and 'other' trials when rats were 'alone' and 'together' during the directional cue epoch (i.e., the 5 s epoch immediately preceding outcome delivery when both outcome and direction were known). Reward and shock indices were significantly shifted in the positive and negative direction for all comparisons (*Figure 2C–F*; Wilcoxon; p's < 0.05). Overall, these results demonstrate that rats understood cues and predicted future outcomes, with reward and shock trials eliciting higher and lower percent beam breaks into the food cup during 'self' trials, respectively. Notably, this pattern of beam breaks between trial types was also present during 'other' trials, but to a lesser degree (*Figure 2B and G–J*).

## Beam breaks into the food cup are modulated by social context

In the above section, we describe how cues that predicted reward and shock increased and decreased beam breaks into the food cup on reward and shock trials, respectively. Interestingly, the

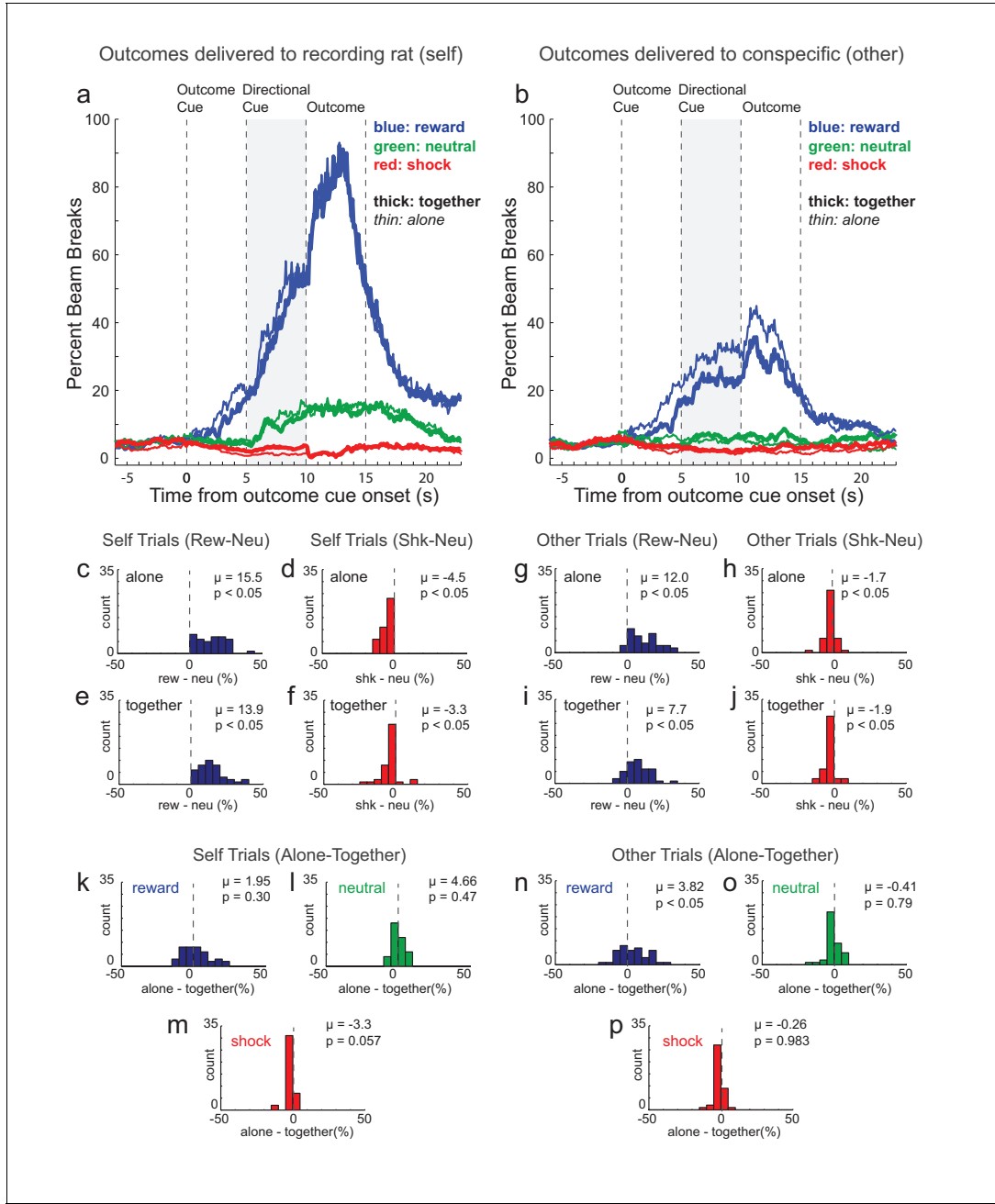

**Figure 2.** Beam breaks in food cup. An infrared beam was placed above the recording rat's food cup. (A) and (B) illustrate percent bream breaks over trial time during 'self' and 'other' trials, respectively (n = 40 sessions; 8 rats). (C–J) During the directional cue epoch (gray) of each session indices were computed by subtracting percent beam breaks during neutral trials from percent beam breaks on shock (red) and reward (blue) trials during 'self' (C-F; left panels) and 'other' trials (G-J; right panels) when rats were 'alone' (C,D,G,H) or 'together' (E,F,I,J). (K–P) During the directional cue epoch of each session indices were computed by subtracting percent beam breaks during 'together' trials from 'alone' trials for reward (blue), neutral (green), and shock (red) trials when the outcome was to be delivered to 'self' (left panels) or 'other' (right panels). Distributions of indices were deemed significantly shifted from zero via Wilcoxon (insets provide mean (μ) and p value).

DOI: https://doi.org/10.7554/eLife.38090.003

divergence in beam breaks on reward and shock trials (relative to neutral) was slightly amplified when rats were alone (*Figure 2A,B*). That is, food cup entries tended to be higher and lower on reward (blue) and shock (red) trials when rats were alone (thin) and together (thick), respectively, with no apparent differences between alone and together neutral (green) trials. This was most

pronounced during reward-other trials, where beam breaks were higher when rats were alone versus together. More interestingly, for shock-self trials (*Figure 2A*; thick red line versus thin red line), food cup entries tended to be more continuously suppressed on 'alone' trials when measured from pre- sentation of the shock cue until delivery of the shock, at which point, alone-shock-self and together- shock-self trials converged. To quantify differences between 'alone' and 'together' trials we plotted distributions of difference scores for beam breaks subtracting 'together' from 'alone' trials for each trial-type (reward, neutral, and shock) during the directional cue epoch (*Figure 2K–P*).

For reward-self trials this distribution was not significantly shifted (*Figure 2K*; Wilcoxon; u = 0.20; p = 0.30), however the distribution of indices were shifted in the positive direction for reward-other trials (*Figure 2N*; Wilcoxon; u = 0.38; p < 0.05). Thus, rats visited the food cup significantly more often on reward-other trials when they were alone. For neutral trials, beam breaks did not differ between 'alone' and 'together' trials (*Figure 2L and O*; Wilcoxon; p's > 0.47). For shock-self, the dis- tribution of 'alone minus together' indices were shifted in the negative direction, almost reaching significance (*Figure 2M*; Wilcoxon; u = −0.09; p = 0.057), possibly reflecting reduced fear when together with the other rat. Notably this effect cannot reflect an overall increase in food cup entries when rats were together because beam breaks on alone and together trials were not significantly lower on alone trials during neutral trials (*Figure 2L and O*), reward trials (*Figure 2K and N*) or shock-other trials (*Figure 2P*).

## Freezing and approach were modulated by social context

Above we show that rats understand the meaning of cues and exhibit increases and suppression of beam breaks in the food cup during reward and shock trials relative to neutral trials, respectively. Here, we scored video to determine the reactions of the recording rat to cues that predicted self- versus other- shock. Although rats froze similarly on together-shock-self and together-shock-other trial types, our data demonstrate that rats froze more to cues that predicted self-shock compared to cues that predicted shock to the conspecific and that rats did not freeze during conspecific shock compared to neutral trials. Further, rats approach the conspecific during shock delivery when the shock was directed at the recording rat more than when the shock was administered to the conspecific.

*Figure 3A* represents average freezing during the directional cue and outcome epochs (5 s each), respectively. Freezing was defined as the absence of movement except for respiration. As for the analysis of beam breaks into the food cup, it was clear that rats understood the meaning of the cues; rats froze significantly more often on shock-self trials compared to other trial types during the directional light epoch, both when together and alone (Wilcoxon; p < 0.05). Also, consistent with the suppression observed on alone-shock-self trials, rats froze significantly more on alone-shock-self trials compared to together-shock-other trials (*Figure 3A*; red; left versus right panels), suggesting less fear during cues that predicted self-shock when rats were together. These effects on freezing were absent during 'other' trials; significant increases in freezing were not observed on shock trials compared to neutral trials during directional cue and outcome epochs (*Figure 3C and D*; Wilcoxons; p's > 0.05).

Thus, under our experimental conditions, rats did not overly exhibit freezing behavior during observation of another rat freezing during cues that predict shock or observation of shock itself, at least when comparing freezing during together-shock-other trials to alone-shock-other and neutral- other trials. However, freezing between together-shock-self and together-shock-other comparisons (i.e., red bar in *Figure 3A* with the open red bar in *Figure 3C*) showed that rats freeze similarly to self-shock and conspecific-shock, consistent with the existing literature. To further quantify this effect we ran a 3-factor ANOVA (together vs alone; self vs other; trial-type (i.e., reward, neutral, shock) on freezing during the directional cue. Significant main effects of together/alone (F(1,396) = 4.44; p < 0.05) and trial-type (F(2,396) = 16.48; p < 0.05), as well as a significant interaction between self/ other and trial-type (F(2,396) = 3.3; p < 0.05) were observed. Thus, rats froze more on shock trials when alone compared to when they were together. Furthermore, post-hoc *t* tests demonstrate that freezing was significantly stronger on self-shock trials when alone compared to when rats were together (t(33) = 2.54; p < 0.05) and that freezing did not significantly differ on together-self-shock and together-other-shock trials (t(33) = 0.19; p = 0.85).

Along with freezing we scored approach. Approach was defined as the movement of the record- ing rat in the direction of the mesh divider, which has been suggested to be a measure of attention,

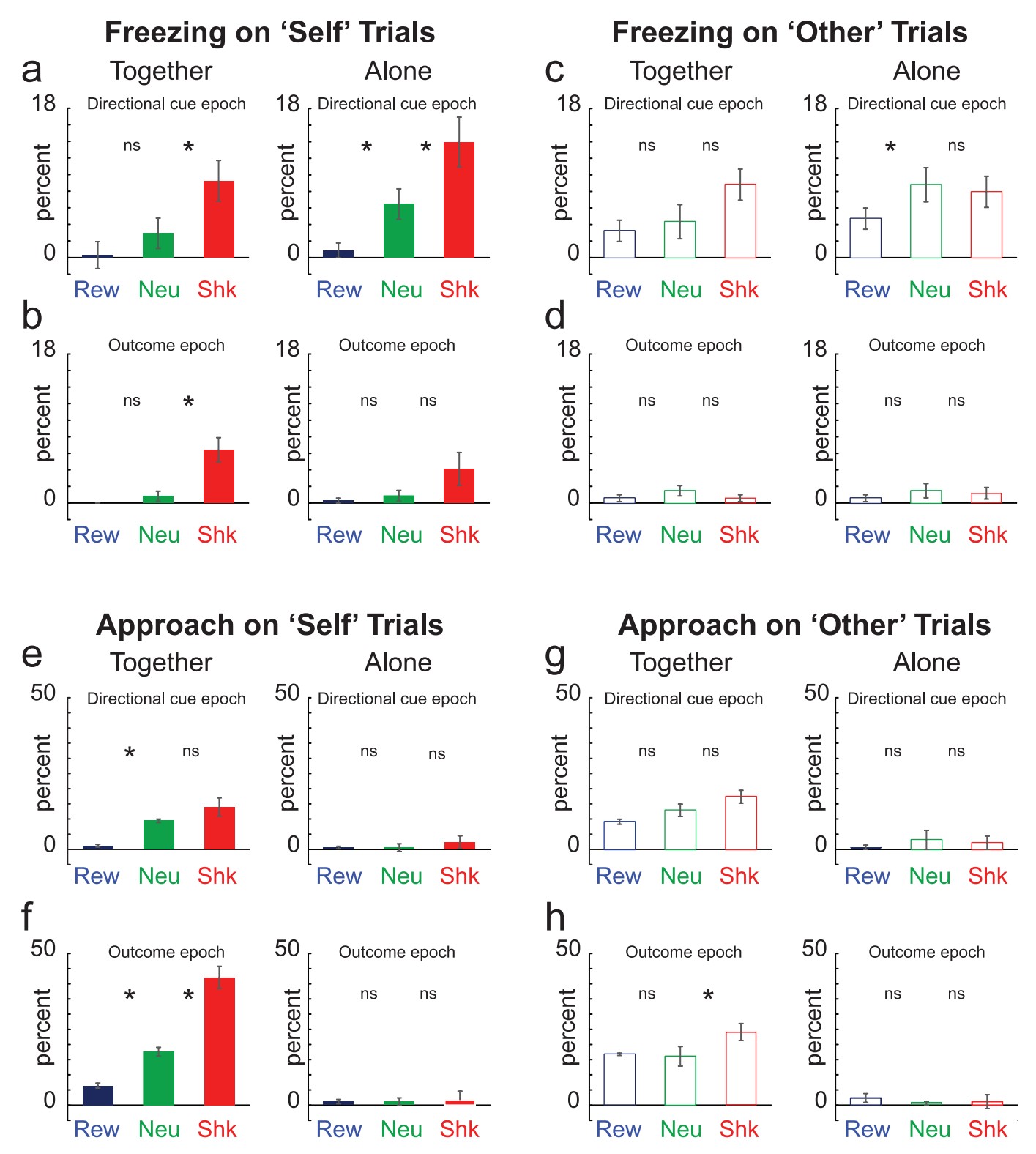

**Figure 3.** Freezing and Approach. (**A–D**) Percent freezing over 12 trial types for 'self' (A,B; solid) and 'other' (C,D; open) trials during the 5 s directional cue epoch (A,C) and the 5 s outcome epoch (B,D) when rats were 'alone' (left) and 'together' (right). Blue = Reward; Green = Neutral; Red = Shock. (**E–G**) Contingencies same as A-D except for percent approach. Note that n = 34 instead to 40 due to technical issues during video recording. *Wilcoxon; p < 0.05.

DOI: https://doi.org/10.7554/eLife.38090.004

concern, and consolation opportunity (*Atsak et al., 2011*; *Ben-Ami Bartal et al., 2011*; *Ben-Ami Bartal et al., 2014*; *Burkett et al., 2016*; *Meyza et al., 2017*). Rats approached the conspecific's side of the box most often on together-shock-self trials (*Figure 3F*; red). For both 'self' and 'other' trial-types, percent approach was significantly higher on together-shock trials compared to alone-shock-self and together-neutral-self trials (*Figure 3F and H*; Wilcoxon; p's < 0.05). Although the recording rat approached the conspecific both during self and other shock trials, this effect was stronger when the threat was to the recording rat (together-shock-self versus together-shock-other; *Figure 3F* versus *Figure 3H*; red; Wilcoxon; p < 0.05). These results suggest that approach was driven by the desire for social contact (i.e., more approach when the conspecific was present than when alone) and shock (i.e., more approach during shock trials than reward and neutral), but only when the recording rat was threatened (i.e., more approach during together-self-shock relative to together-shock-other trials).

## Suppression of DA release during cues that predict shock is reduced when together

The above results demonstrate that rats understand when and who will be shocked, display a general tendency to approach the conspecific when shock is to occur to oneself, and are less fearful during shock-self trials when paired with a conspecific. Next, we examined the impact of social context on DA release. It has previously been shown that cues that predict appetitive events increase DA firing and release, whereas cues that predict aversive events suppress firing and DA release (*Bromberg-Martin et al., 2010*; *McCutcheon et al., 2012*; *Oleson et al., 2012*; *Roesch et al., 2010*; *Roitman et al., 2008*; *Schultz, 2010*; *Schultz et al., 2015*; *Schultz et al., 1997*; Wenzel et al., 2018). Consistent with this previous work, we found that when rats were alone, average DA release over all sessions (n = 40) increased to cues that predicted reward-self trials (blue) significantly more than neutral trials (green) during presentation of cues and reward (*Figure 4A*) and that DA release was lower during presentation of cues that predicted unavoidable shock (*Figure 4A*; red (shock) versus green (neutral) when alone (thin) during directional light cue presentations). This is further illustrated and quantified in *Figure 4F and G*, which plots the distribution of reward indices (reward minus neutral trials) and shock indices (shock minus neutral trials) for DA release during the directional cue epoch during 'alone' trials for each session. The distributions of reward and shock indices were significantly shifted in the positive and negative direction indicating significantly higher (*Figure 4F*; Wilcoxon; μ = 24.6; p < 0.05) and lower (*Figure 4G*; Wilcoxon; μ = −11.5; p < 0.05) DA release on reward and shock trials relative to neutral trials, respectively.

Thus, as described previously, DA release increased and decreased during cues that predicted appetitive and aversive events when presented to the recording rat (*Bromberg-Martin et al., 2010*; *McCutcheon et al., 2012*; *Oleson et al., 2012*; *Roesch et al., 2010*; *Roitman et al., 2008*; *Schultz, 2010*; *Schultz et al., 2015*; *Schultz et al., 1997*; *Wenzel et al., 2018*). Next, we asked if signals during 'self' trials were modulated by social context. During reward-self trials, there were no differences in DA signaling between 'alone' and 'together' trials (*Figure 4A*; blue thin vs. thick lines). However, during shock-self trials, we found that the decline of DA release induced by the presentation of shock-self cues was attenuated in the presence of the conspecific (*Figure 4A*; thick versus thin red lines after outcome cue onset). Consistent with this observation, the distribution of 'shock minus neutral' indices calculated on shock-self trials were not significantly shifted (*Figure 4I*; Wilcoxon; μ = −1.4; p = 0.43) as they were during alone-shock-self trials (*Figure 4G*; Wilcoxon; μ = −11.5; p < 0.05).

To directly quantify differences between 'alone' and 'together' trials, we computed an index for each session that subtracted DA release on 'together' trials from DA release on 'alone' trials during the directional light cue epoch. Distributions of indices for DA release during the directional cue epoch were significantly shifted in the negative direction during shock trials (*Figure 4D*; μ = −12.1; p < 0.05), but not for reward (*Figure 4B*; μ = −0.4; p = 0.86) nor neutral trials (*Figure 4C*; μ = −2.0; p = 0.25), demonstrating that in most sessions, DA release was significantly lower during alone-shock-self trials compared to together-shock-self trials. The box and whisker plots in *Figure 4E* illustrate these effects across sessions (small dots) and rats (large dots) color coded by rat identity and regressions between alone and together for beam breaks and DA release are illustrated in *Figure 4—figure supplement 3*.

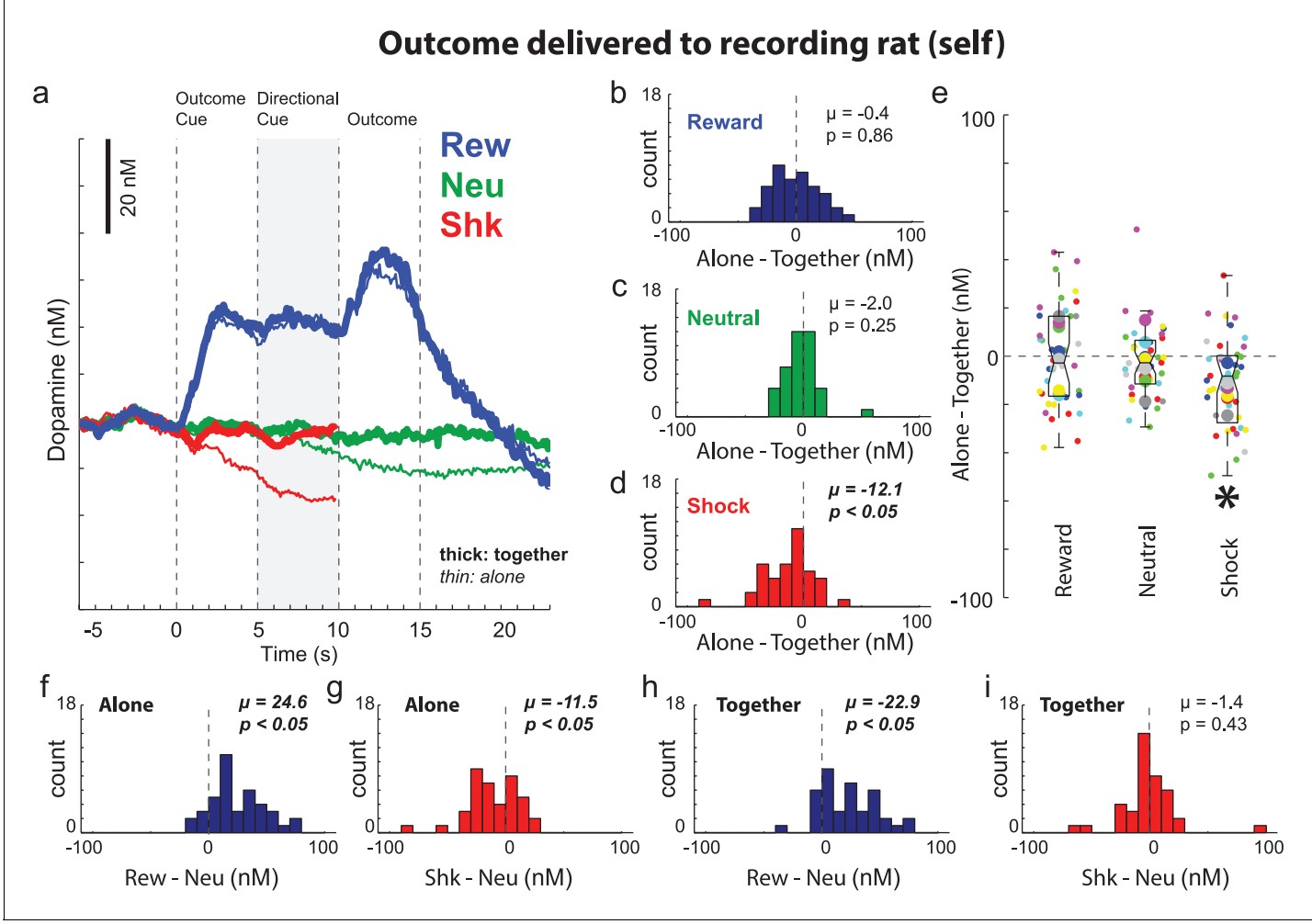

**Figure 4.** Dopamine release when the outcome was delivered to the recording rat (i.e., self). (A) Average DA release over trial time (n = 40 sessions, 8 rats). See *Figure 4—figure supplements 1* and *2* for example sessions. Shock trials (red) are truncated due to shock artifact. (B–D) During the directional cue epoch (i.e., 5 s after directional cue) of each session indices were computed by subtracting average DA release during 'together' trials from 'alone' trials (Alone minus Together) for reward (blue; B), neutral (green; C), and shock (red; D) during 'self' trials. (E) Distributions of the same indices as in B-D (Alone – Together for reward, neutral and shock trials) shown by session (small dots) and rat (large dots) color coded by rat identity. See *Figure 4—figure supplement 3* for regressions between behavior and DA release by session and rat. (F–I) Reward (reward – neutral) and shock (shock – neutral) indices taken during the directional cue epoch when rats were alone (F and G) or together (H and I). Distributions of indices were deemed significantly shifted from zero via Wilcoxon (insets provide mean (μ) and p value).

DOI: https://doi.org/10.7554/eLife.38090.005

The following figure supplements are available for figure 4:

**Figure supplement 1.** Example false-color plots for reward-self, reward-other, and shock-other trial-types.
DOI: https://doi.org/10.7554/eLife.38090.006

**Figure supplement 2.** Example false-color plots for shock-self and shock-other trial-types when together and alone.
DOI: https://doi.org/10.7554/eLife.38090.007

**Figure supplement 3.** Regressions between alone and together for beam breaks and DA release during shock-self trials.
DOI: https://doi.org/10.7554/eLife.38090.008

Finally, it is important to note that although we observed no differences between 'alone' and 'together' neutral trials during the directional light cue epoch (*Figure 4C*; Wilcoxon; p = 0.25), differences between alone and together neural trended at the end of the outcome epoch, nearly obtaining significance (Wilcoxon; p = 0.06). This result may reflect social engagement that occurs on neutral trials independent of other trial manipulations, as reported previously (*Robinson et al., 2003*; *Willuhn et al., 2014*).

## Dopamine release increased when the conspecific was shocked

We have shown that DA release increases and decreases to cues that predict reward and shock to oneself, respectively. Here we ask if similar signals were observed when rewards and shocks are directed to the conspecific. Average DA release during trials in which outcomes were delivered to the conspecific rat (i.e., 'other' trials) is shown in *Figure 5A*. After the cue signaled a reward trial, there was an increase in DA release above neutral cues similar to what was observed during reward-self trials (Wilcoxon; p < 0.05). Recall that during this epoch (i.e. before the directional cue) rats were unaware of who would receive the outcome. Thus, prior to cue onset to the directional cue, increases in DA release might simply reflect the associated value of possibly receiving reward for oneself (50/50), not the value of reward associated with reward being delivered to the conspecific.

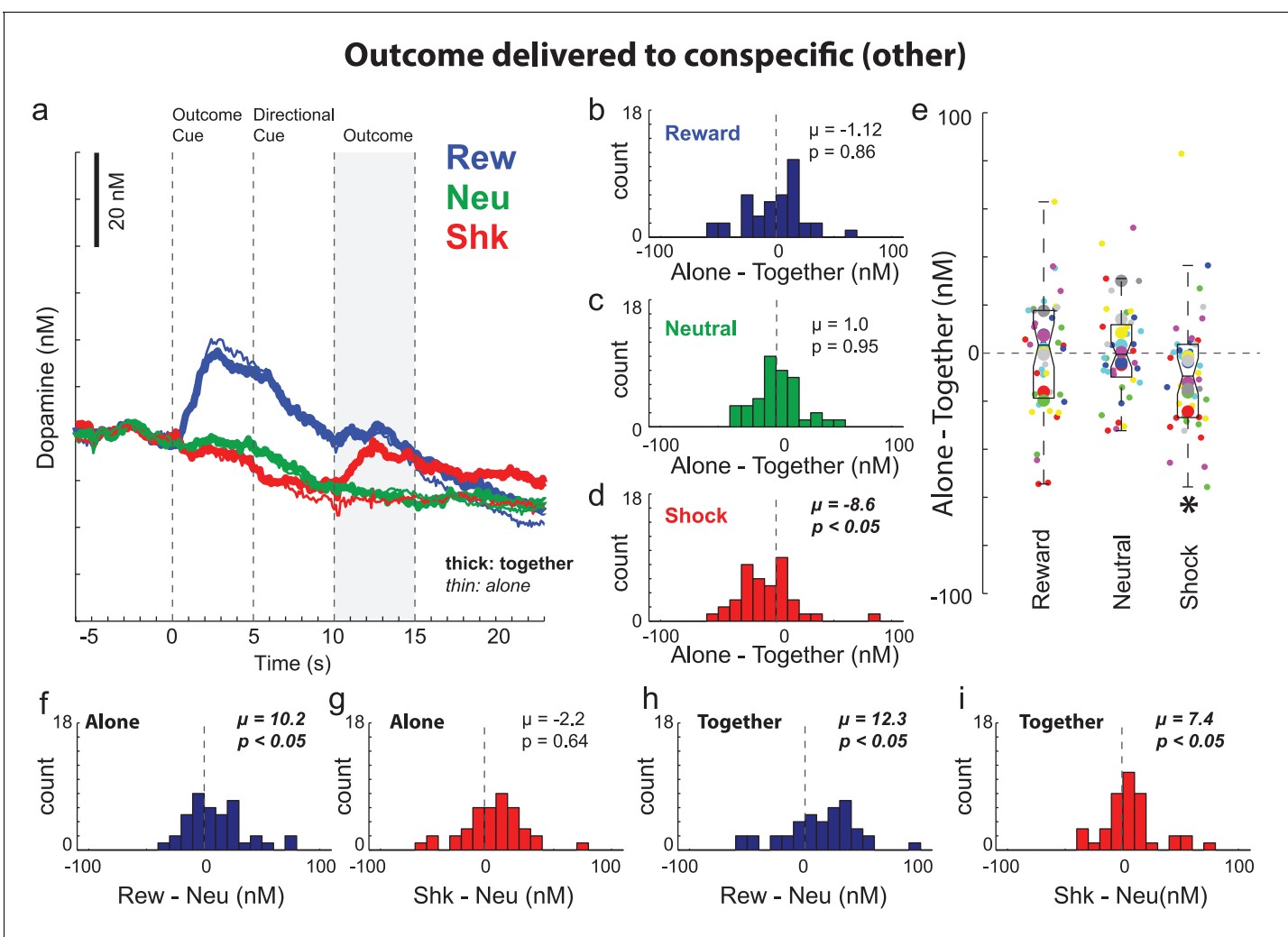

**Figure 5.** Dopamine release when the outcome was delivered to the conspecific (i.e., other). (A) Average DA release over trial time (n = 40 sessions, 8 rats). See *Figure 4—figure supplements 1* and *2* for example sessions. (B–D) During the directional cue epoch of each session indices were computed by subtracting average DA release during 'together' trials from 'alone' trials (Alone minus Together) for reward (blue; B), neutral (green; C), and shock (red; D) during 'other' trials. (E) Distributions of the same indices as in B-D (Alone – Together for reward, neutral and shock trials) except shown by session (small dots) and rat (large dots) color coded by rat identity. *Figure 5—figure supplement 1* for regressions between behavior and DA release by session and rat. (F–I) Reward (reward – neutral) and shock (shock – neutral) indices taken during the directional cue epoch when rats were alone (F and G) or together (H and I). Distributions of indices were deemed significantly shifted from zero via Wilcoxon (insets provide mean (μ) and p-value).
DOI: https://doi.org/10.7554/eLife.38090.009

The following figure supplement is available for figure 5:

**Figure supplement 1.** Regressions between alone and together for beam breaks and DA release during shock-other trials.
DOI: https://doi.org/10.7554/eLife.38090.010

Only after directional cue presentation did rats become aware of who would receive the reward. In line with rats being self-interested, immediately after illumination of the directional light cue, DA release declined to pre-cue levels for both 'alone' and 'together' trials. As a result, DA release was significantly lower when reward was delivered to the conspecific compared to when reward was delivered to the recording rat (Wilcoxon; $p < 0.05$). This was true for both 'alone' and 'together' trials and there were no significant differences between them during the outcome epoch (*Figure 5B*; Wilcoxon; $\mu = -1.12$; $p = 0.86$), which suggest that DA release during reward-other trials was not modulated by social context. Arguably, this suggests that reward delivered to the conspecific was not better than reward delivered to the empty box. Furthermore, the relative decrease in DA release after onset of the directional cue suggests that recording rats may have inferred that they were not going to receive a reward during the trial, and that this was independent from whether or not the recording rat or conspecific received (or did not receive) a reward.

Although DA release on reward-other trials was not modulated by presence of the conspecific, shock-other trials were. Prior to shock delivery DA release was similar during the outcome cue and directional cue epochs (*Figure 5A*). In fact, DA release was not different on shock-other trials compared to together-shock-self trials, reminiscent of the observation described above that rats froze at a similar level on together-shock-other compared to together-shock-self trials. To further quantify these effects we performed a three factor ANOVA (together vs alone; self vs other; trial-type (ie, reward, neutral, shock) on DA release during the directional cue epoch. We found main effects of self/other ($F_{(1,468)} = 17.53$; $p < 0.05$) and outcome ($F_{(1,468)} = 78.25$; $p < 0.05$), and an interaction between self/other and outcome ($F_{(2,468)} = 5.14$). Post-hoc t-tests yield a significant difference between alone-shock-self and together-shock-self ($t_{(39)} = 3.53$; $p < 0.05$) consistent with the analysis described above, and no significant difference between together-self-shock and together-conspecific-shock trials ($t_{(39)} = 1.95$; $p = 0.06$).

Although prior to shock delivery, DA release was similar for together-shock-self, together-shock-other and alone-shock-other trials, we found that after shock was delivered to the conspecific, DA release actually increased (*Figure 5A*; thick red); DA release was significantly higher following shock delivery to the conspecific compared to neutral trials (*Figure 5A*; thick red versus thick green; *Figure 5I*; Wilcoxon; $\mu = 7.4$; $p < 0.05$) and compared to when shock was delivered to the conspecific's empty side of the cage (*Figure 5A*; thick red versus thin red; *Figure 5D*; Wilcoxon; $\mu = -8.6$; $p < 0.05$). *Figure 5E* illustrates these effects across sessions (small dots) and rats (large dots) color coded by rat and regressions between alone and together for beam breaks and DA release are illustrated in *Figure 5—figure supplement 1*.

## Instrumental social distress task

Taken together the analysis of behavior and DA release during the Pavlovian Social Distress Paradigm suggests that although rats show signs of empathetic/pro-social behaviors (*Atsak et al., 2011*; *Ben-Ami Bartal et al., 2011*; *Ben-Ami Bartal et al., 2014*; *Burkett et al., 2016*; *Meyza et al., 2017*), many of their behaviors could also be characterized as 'self-interested' and DA release better reflected the subjective valuation of events as opposed to the objective value of stimuli and outcomes. However, since that paradigm was completely Pavlovian, rats were never given the opportunity to perform 'prosocial' acts that relieve distress for others, making it difficult to truly characterize the DA response and the nature of their behavior. That is, in the Pavlovian task described above, the recording rat had no control over whether or not the conspecific would be shocked.

To address this issue, we had a different cohort of rats (n = 9; *Figure 6H*) perform an Instrumental Social Distress Task, where, on a portion of trials, performing an instrumental act (i.e., lever press) to obtain reward for oneself also resulted in a shock to the conspecific. The task is illustrated in *Figure 6*. On each trial, rats were presented with a lever 5 s after presentation of an auditory stimulus. On some of these trials the auditory stimulus predicted either shock to oneself (shock-self; red; *Figure 6A,C*) or to the conspecific (shock-other; orange *Figure 6A,D*) upon lever press. Each of these occurred randomly on 12.5% of trials. On the remaining 75% of trials, the auditory stimulus signaled that a lever press would not produce shock (*Figure 6A,B*; no-shock; blue). For all trial types the same reward (sucrose pellet) was delivered after the lever press. After extension of the lever, rats had 5 s to press, otherwise the lever retracted, and no reward or shock was delivered. On shock-self trials (*Figure 6A,C*; red) and shock-other trials (*Figure 6A,D*; orange), if the recording rat

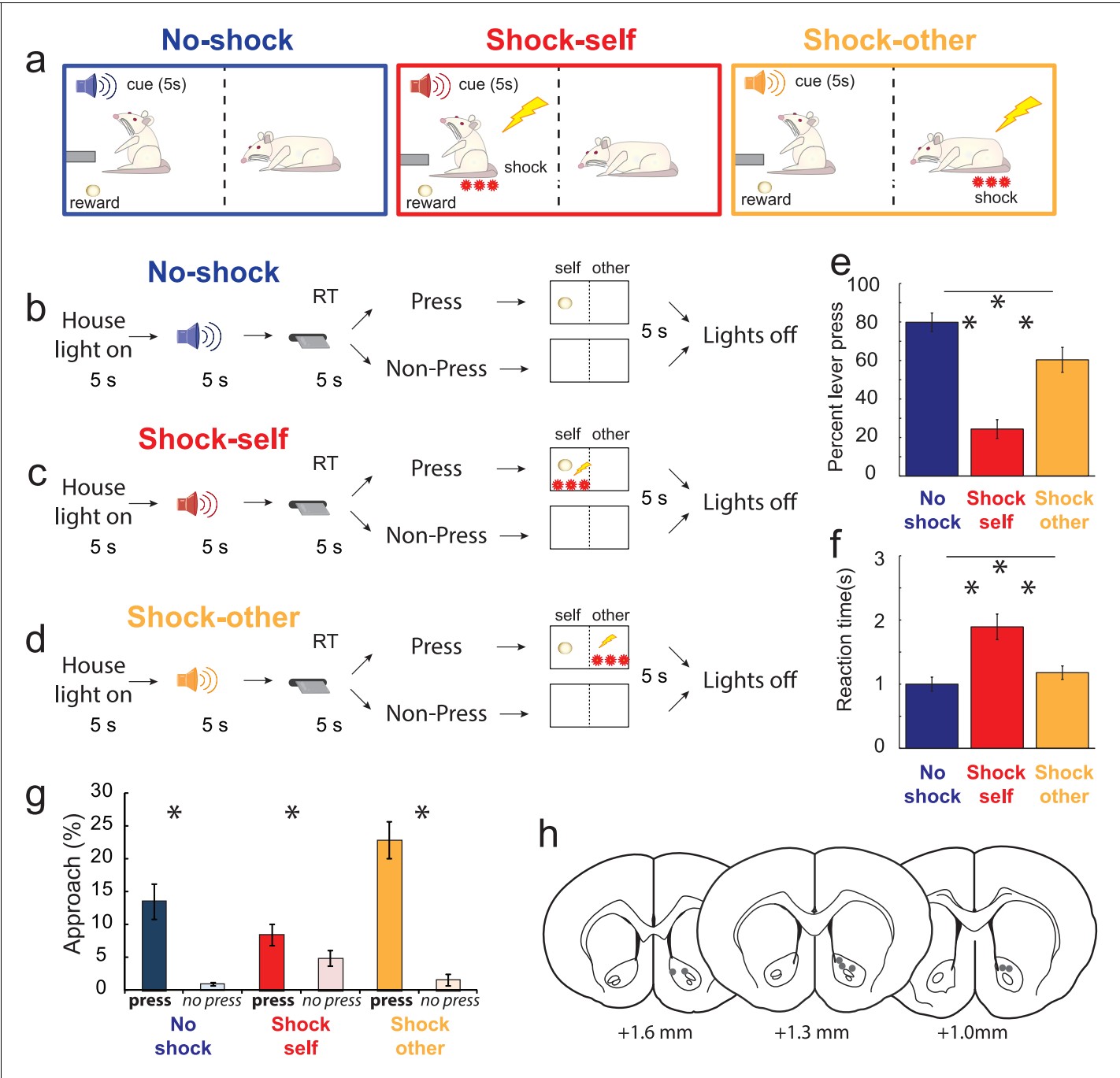

**Figure 6.** Instrumental Social Distress Task. (**A–D**). Rats performed a task during which three independent auditory stimuli predicted three different trial-types. All lever presses led to one sucrose pellet delivery. One stimulus signaled that a subsequent lever press would result in reward with no shock (**A, B**; 'no-shock', blue). The second auditory stimulus signaled that the lever press would produce shock to oneself along with delivery of reward (**A,C**; 'shock-self'; red). The third stimulus predicted shock to the conspecific upon lever press and reward delivery to the lever-presser (**A,D**, 'shock-other'; orange). (**E** and **F**) Percent lever press and reaction time (lever out to lever press) over 24 sessions (n = 8 rats). (**G**) Percent approach by the recording rat toward the conspecific after extension of the lever to offset of lights. (**H**) Placement of chronic recording electrodes based on histology. *Wilcoxon; p < 0.05.

DOI: https://doi.org/10.7554/eLife.38090.011

pressed the lever, reward was delivered to the recording rat along with a shock to either the recording rat or the conspecific.

## Rat behavior during the instrumental social distress task

Over all recording sessions (n = 24), rats took longer (*Figure 6F*) and were less likely (*Figure 1E*) to lever press on shock-other trials relative to non-shock trials (Wilcoxon; p < 0.05) consistent with previous reports suggesting thats rat do perform 'prosocial' acts to benefit others in distress (*Atsak et al., 2011*; *Ben-Ami Bartal et al., 2011*; *Ben-Ami Bartal et al., 2014*; *Burkett et al., 2016*; *Meyza et al., 2017*). However, behavior was also strongly guided by individual outcomes rather than those of the conspecific. That is, rats refrained from pressing significantly more often and were significantly slower when the auditory cues signaled shock to oneself as opposed to the conspecific (*Figure 6E and F*; red vs orange; Wilcoxon; p's < 0.05).

The results illustrate that rats can integrate potential harm into their decision-making and that they differentiate between auditory stimuli predicting shock to oneself and to the conspecific. Furthermore, based on our data, rats likely were aware that lever pressing could lead to aversive consequences to the conspecific. On footshock trials, rats approached the conspecific on 24% of trials, but only following the lever press (*Figure 6G*; orange; 23% after lever press; 1% after no lever press; chi-square, p < 0.05). Interestingly, rats also occasionally approached the conspecific after the lever press on no-shock trials (*Figure 6G*; blue; 13% after lever press; 1% after no lever press; chi-square, p < 0.05). Altogether, these results suggest that the recording rats differentiated between trial types and that their behavior led to conspecific distress as measured by reduced lever pressing on shock-other trials compared to no-shock reward trials, as well as increased approach after lever press trials. The results demonstrate that rats do exhibit pro-social behavior as described previously (*Atsak et al., 2011*; *Ben-Ami Bartal et al., 2011*; *Ben-Ami Bartal et al., 2014*; *Burkett et al., 2016*; *Meyza et al., 2017*), but also indicate that the recording rat placed more value on self-preservation as opposed to sparing the conspecific.

## Cue-evoked DA release was lower during shock-other trials

Although cues that predict unavoidable footshock suppress DA release (as in the Pavlovian Social Distress Paradigm described above; *Figure 4*), recent work has shown that cues that predict avoidable shock and avoidance of shock itself, DA release increases to a similar degree as to cues that predict reward delivery (*Gentry et al., 2016*; *Oleson and Cheer, 2013*; *Oleson et al., 2012*; *Wenzel et al., 2018*). Here we have replicated those results. *Figure 7* illustrates DA release over all recording sessions (n = 24) when rats pressed for reward (blue) and did not press to avoid self-shock (red). DA release was observed on both reward and shock-self cues, with no difference between them. This is quantified in *Figure 7C*, which plots the difference between DA release on reward press and shock-self no press trials (shock minus reward) for each session. The distribution is not significantly shifted indicating no differences between reward and shock-self trials (Wilcoxon; μ = −1.9; p = 0.95). Thus, unlike cues that predict unavoidable shock, which are aversive and suppress DA release (*Figure 4*), cues that predict avoidable shock increase DA release, reflecting the value of avoiding an aversive outcome in negative reinforcement. Next, we asked if DA release was equally high when recording rats spared the conspecific.

Orange lines in *Figure 7A* represent shock-other trials when the recording rat did not press the lever for reward, thus avoiding shock for the conspecific. Remarkably a similar pattern of DA release emerged as during non-press shock-self trials, with DA release to the cue and during the absence of the shock; however, the magnitude of DA release was significantly lower on shock-other no press trials (orange) compared to when rats avoided self-shock (red) and when the rats pressed the lever on no-shock reward trials when there was no threat of shock (blue). This is quantified in *Figure 7B and D*, which plot the distributions of differences between shock-other and shock-self (shock-self minus shock-other), and between no-shock and shock-other trials (shock-other minus no-shock). Both distributions were significantly shifted (Wilcoxon; p's < 0.05; *Figure 7B and D*) indicating significantly lower DA release on shock-other trials relative to no-shock trials and significantly higher DA release on shock-self relative to shock-other trials. Thus, although cues that signal avoidable shock and shock avoidance increased DA on 'other' trials, the strength of this signal was weaker than cues that predicted avoidable shock or reward for the recording rat. *Figure 7E* illustrates these effects across

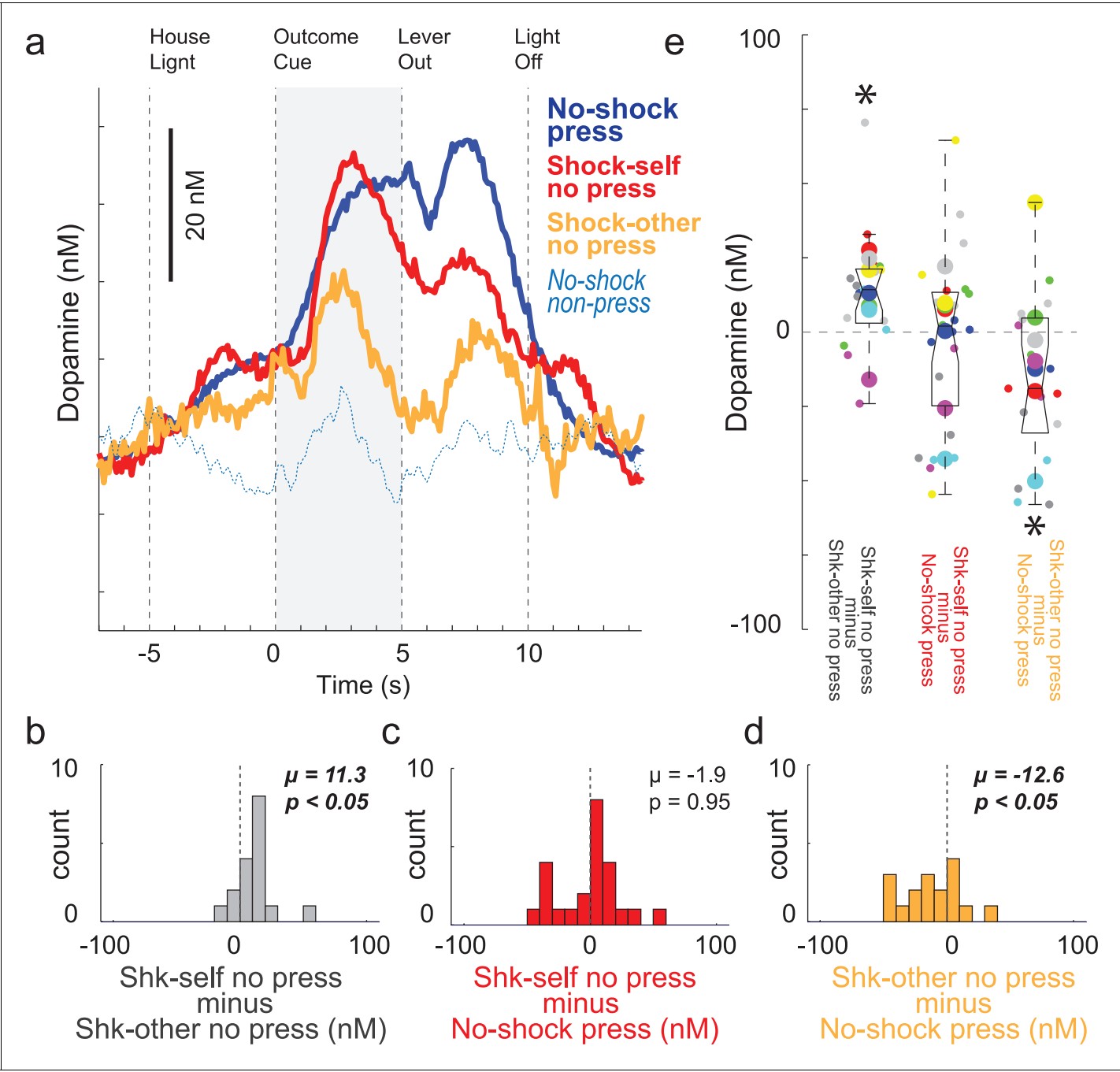

**Figure 7.** DA release during performance of the Instrumental Social Distress Task. (**A**) Average DA release (n = 24 sessions, 9 rats) over time for no-shock press trials (blue), shock-self no press trials (red), shock-other no press trials (orange), and no-shock non-press trials (light blue dashed). See *Figure 7—figure supplement 1* for an example session. Note that trials during which the recording rats were shocked could not be shown due to significant noise in the signal and that there were no significant differences between shock-other press and no-shock press trials during the reward epoch (ttest; p = 0.08). (**B–D**) During the outcome cue epoch (5 s after cue onset) indices comparing DA release between trial-types were computed for each session. Distributions of these indices are shown in B-D. B = Shk self no press minus Shk-other no press; C = Shk self no press minus No-shock press; D = Shk other no press minus No-shock press. (**E**) Distributions of the same indices as in B-D except shown by session (small dots) and rat (large dots) color coded by rat identity. See *Figure 7—figure supplement 2* for regression between behavior and DA release by sessions and rat. Distributions of indices were deemed significantly shifted from zero via Wilcoxon (insets provide mean (µ) and p-value).

DOI: https://doi.org/10.7554/eLife.38090.012

The following figure supplements are available for figure 7:

**Figure supplement 1.** Example false-color plots for no-shock press, shock-self no press, and shock-other no press trial-types.

*Figure 7 continued on next page*

*Figure 7 continued*

DOI: https://doi.org/10.7554/eLife.38090.013

**Figure supplement 2.** Regressions between 'self' and 'other' for behavior and DA release.

DOI: https://doi.org/10.7554/eLife.38090.014

sessions (small dots) and rats (large dots) color coded by rat and regressions between 'self' and 'other' for behavior and DA release are illustrated in *Figure 7—figure supplement 2*.

## Discussion

This study is significant given the absence of work examining modulation of neural signals in social tasks where conspecific distress impacts behavior and decision making. Importantly, patients with psychopathic traits and autism have difficulties recognizing the mental states of others and appropriately altering behavior in social contexts (*Baron-Cohen et al., 1985*; *Blair, 2003*; *Buckholtz et al., 2010*; *Cook and Black, 2012*; *Gaigg, 2012*; *Hamilton, 2013*; *Matthys et al., 2013*; *Neuhaus et al., 2010*; *Rizzolatti and Fabbri-Destro, 2008*; *Scott-Van Zeeland et al., 2010*; *Taylor and DeQuinzio, 2012*). These disorders are difficult to treat and currently lack effective pharmacological interventions, reflecting the lack of knowledge pertaining to our understanding of the underlying neurobiology of how social cues may alter behavior.

Although empathy and prosocial behavior might be considered a trait unique to primates, a number of recent studies suggest that rats can display similar behaviors (*Atsak et al., 2011*; *Ben-Ami Bartal et al., 2011*; *Ben-Ami Bartal et al., 2014*; *Burkett et al., 2016*; *Church, 1959*; *de Lecea et al., 2006*; *Guzmán et al., 2009*; *Hernandez-Lallement et al., 2018*; *Hernandez-Lallement et al., 2016*; *Hernandez-Lallement et al., 2017*; *Jeon et al., 2010*; *Jones et al., 2014*; *Kashtelyan et al., 2014*; *Kim et al., 2010*; *Langford et al., 2006*; *Masserman et al., 1964*; *Mogil, 2015*; *Panksepp, 2011*; *Panksepp and Lahvis, 2011*; *Sato et al., 2015*). For example, rats will choose mutual reward in a prosocial task, freeze when they see another rat freeze (interpreted as 'empathy') and exhibit 'prosocial' rescue behavior of a conspecific in distress. Most notable was a recent set of high profile papers (*Science; eLife*) demonstrating that rats will rescue a conspecific that is confined in a restrainer, even when faced with a choice between freeing the trapped rat and obtaining food reward (*Ben-Ami Bartal et al., 2011*; *Ben-Ami Bartal et al., 2014*). Although many of these studies are intriguing, they have met criticism; effects are often small, rats only freeze when they have experienced shock themselves, and rats do not free trapped conspecifics under all circumstances (*Schwartz et al., 2017*; *Silberberg et al., 2014*). Further, many of these effects are dependent on rats being cage mates or of the same strain, suggesting that 'prosocial' rescue behavior might reflect some sort of reproductive benefit.

It is well-known that activity of DA neurons and DA release in NAc increases and decreases to events that are better and worse than expected, respectively (*Bayer and Glimcher, 2005*; *Bromberg-Martin et al., 2010*; *Day et al., 2007*; *Gan et al., 2010*; *Hart et al., 2015*; *Hart et al., 2014*; *McCutcheon et al., 2012*; *Roesch et al., 2010*; *Schultz, 2010*; *Schultz et al., 2015*; *Schultz et al., 1997*). This is true across species, techniques and tasks. Indeed, in our task, reward delivery and cues that predicted reward delivery, increased DA release when they were directed to the recording rat. Similarly, cues that predicted unavoidable shock inhibited DA release when it was the recording rat that was to be shocked. Importantly, cues and outcomes directed to the conspecific did not produce the same pattern of results, suggesting that – if DA signals genuinely reflect value of predictive cues – cues that predicted conspecific reward and shock were not considered to be appetitive and aversive, respectively. Moreover, footshocks directed to the conspecific actually increased DA release, suggesting that rats may have been using cues from conspecific (e.g., vocalizations, odors, visual, etc.) to determine that the shock had been delivered to the conspecific, signaling the value of personal safety or relief. This interpretation is supported by the freezing and approach behavior at the time of the shock. The rat that we recorded from approached the other rat during the outcome, but did not freeze. This suggests that the recording rat was engaged with the conspecific possibly to determine what will occur during the trial, but was not overly fearful during that period (i.e., the period when DA increases during conspecific shock). If rats were more strongly displaying empathy or the objective value of the conspecific shock, DA release would be inhibited to a stronger degree

and rats would freeze more to the conspecific being shocked compared to when a shock is applied to the empty box. This is a very interesting aspect of the data and can only be obtained when trials are anti-correlated. Of note, in previous 'empathy' studies there were no real consequences to the animal when a negative event occurred to the conspecific. Here, it appears that the 'relief' or 'safety' aspect stemming from the situation, through negative reinforcement processes, outweighed any signals that reflect the evaluation of the nature of the outcome from the perspective of the conspecific. We do not believe that DA signals 'relief' or 'empathy' per se, but reflects prediction error correlates that are informed by social contexts. In this case, since the trial types are anti-correlated, DA increases because the rat being recorded from did not get shocked, eliciting a positive prediction error. Nevertheless, our data does not unambiguously demonstrate that the response is due to the rat's own safety and it remains unclear whether we can completely exclude the possibility that this response is due to the outcome presented to the conspecific. Disambiguating this point requires additional control experiments that would require isolating DA signals during conspecific alone when there are no self-shock trials. This might help dissociate signals related to relief/safety from those that represent the conspecific's outcome; however, it is known that rats will not display empathy while watching another rat get shocked unless they too have been shocked. Thus, rats may not internalize 'conspecific outcome' until they have experienced the negative outcome themselves making this a difficult issue to reconcile.

Overall, these results suggest that DA release signals the subjective value placed on events that occur to the conspecific similar to what has been described previously in the appetitive domain (*Dal Monte et al., 2018*; *Kashtelyan et al., 2014*; *Noritake et al., 2018*). Remarkably, we also found that the presence of the other rat altered the subjective value placed on cues that predicted self-shock by reducing fear and the associated suppression of the DA signal during the Pavlovian paradigm. On these trials the rat approached the conspecific, perhaps reflecting 'consolation' seeking behavior similar to what has been described in prairie voles (*Burkett et al., 2016*). The lack of suppression of DA release during this period suggests that the recording rat found the shock cue to be subjectively less aversive in the presence of the conspecific. Indeed, rats entered the food cup and froze less often on together-shock-self trials. Importantly, these results differ from what has been described in prairie voles; that is, a prairie vole that is not in distress will approach another vole that is in distress. In our task, approach was mostly one-sided in that the recording rat approached the conspecific far less on shock-other trials.

In the context of the instrumental task rats pressed significantly less and were slower to press on conspecific-shock trials, demonstrating empathetic and prosocial behaviors consistent with previous reports (*Atsak et al., 2011*; *Ben-Ami Bartal et al., 2011*; *Ben-Ami Bartal et al., 2014*; *Burkett et al., 2016*; *Meyza et al., 2017*). Remarkably, DA release during decisions to give up reward and spare the conspecific from shock mirrored DA release when the rat avoided shock for itself, suggesting that DA release during conspecific shock trials reflected the value of saving the conspecific similar to when it signals the value of saving oneself and that the same signal that encourages avoiding shock for oneself is present when avoiding shock for a conspecific. However, lever pressing increased and DA release decreased during conspecific-shock compared to self-shock, which indicates that rats were more interested in sparing themselves from harm rather than sparing the conspecific. This suggests that the DA signal is modulated by the degree of 'empathy' exhibited by rats in the context of this task. Importantly, as above, we do not think that DA is signaling empathy per se, but represents modulation of prediction errors in this task by empathy, reflecting the subjective value placed on avoiding conspecific shock.

The behavior described here demonstrates that rats do not always demonstrate 'empathetic' and 'prosocial' behavior as described previously or as expected in humans. This may be overly apparent in our tasks where there is a real consequence to performing the prosocial act (i.e., no rewards on that trial), a looming threat that an upcoming trial might result in personal shock, and no alternative reward (i.e., social contact; explore the restrainer). Although rats might not exhibit human-like empathetic or prosocial behavior in our tasks, to be fair to rat-kind, it is unclear how humans would act in similar tasks when no one is watching. Altogether, our results suggest that there might be something unique about previously published work that does not completely generalize across behavioral contexts – that the unique structure of previous experiments promotes stronger empathy and pro-social behavior, especially when there is not much cost to the actor. Importantly, across studies (ours and others) we don't think that these differences reflect problems with task design (ours or others), but

that each task structure promotes different levels of 'pro-social' and 'empathetic' behavior along a continuum, providing insights into social behavior, and how the brain governs behavior and interprets outcomes in social contexts.

It is possible that prosocial and empathetic behavior may have been more prominent in our paradigms if rats were cage mates; however, previous work has shown that rats will help 'trapped strangers' by releasing them from the restrainer, just as they do for cage mates so long as they were of the same strain (Ben-Ami Bartal et al., 2014). In our study, rat pairs were of the same strain, and were well acquainted in that they had been shipped together in the same box, were paired together in the recording boxes for the entirety of the experiment, and rat pairs resided next to each other in transparent cages in the animal colony. However, we cannot rule out that physical contact may be necessary to promote stronger empathic and prosocial behavior.

It is important to note that our goal was not to uncover the exact correlates of social behavior for we do not believe the sole purpose of this brain circuit is to make social decisions, rather, the purpose of these studies were to understand how social contexts impact decision making and reinforcement circuits, with the hope that we will uncover the cogs in the greater machinery that contribute to complex behaviors (e.g., empathy, prosocial, egocentric or self-interested) in different social contexts. For the first time, we show that DA release is modulated by social context in at least three ways. First, reductions in DA release during cues that predict foot shock were weaker in the presence of the conspecific, likely due to comfort or distraction of being in the presence of another rat as evidenced by seeking social contact prior to and during foot shock delivery when together, but not alone. Second, DA release during shock trials increased when shock was administered to the conspecific, demonstrating that DA release was modulated by conspecific cues that indicated that the foot shock had occurred on the other side of the box. This result may reflect an external signal that foot shock would not occur to the recording rat. Third, during the instrumental task, cues that predict avoidable foot shock for the conspecific elicited a similar, but weaker DA release compared to cues that predicted avoidable foot shock for oneself. Overall, we conclude that DA release is modulated by social context in that rats use social cues to optimize predictions about their own self-interest, even at the expense of a conspecific, and that correlates related to avoidance are present, albeit weaker, when consequences are directed to another rat as opposed to oneself.

## Materials and methods

### Animals
Thirty-four male Sprague-Dawley rats were obtained at 300–350 g from Charles River Labs. Rats were individually housed on a 12 hr light–dark cycle and tested during the light phase. Experiments were conducted during the day. Water was available *ad libitum* and body weight was maintained at no less than 85% of pre-experimental levels by food restriction (14–15 g of laboratory chow daily in addition to approximately 2.5 g of sucrose pellets (Test Diet) consumed during daily experimental sessions). Each implanted animal was paired with the same conspecific throughout the experiment. Conspecifics were of the same age and sex, were ordered at the same time, and housed next to each other in transparent cages in the animal colony. Rats were not housed in the same cage due to implants. All experiments were approved by the University of Maryland College Park Institutional Animal Care and Use Committee (R-15–34; R-JUL-18–37) under university and NIH guidelines.

### Chronic microelectrode fabrication
Electrodes were constructed according to the methods of Clark et al. (2010). A single carbon fiber (Goodfellow Corporation) was inserted into a 15 mm cut segment of fused silica (Polymicro Technologies) while submerged in isopropyl alcohol. One end of the silica tubing was sealed with a two-part epoxy (T-QS12 Epoxy, Super Glue) and left to dry overnight, leaving untouched carbon fiber extending past the seal. The protruding carbon fiber was cut to a length of 150 μm. A silver connector (Newark) was secured to the carbon fiber at the opposing end of the silica tubing using silver epoxy (MG Chemicals) and was allowed to dry. A final coat of two-part epoxy was then applied to the pin connection to provide insulation and structural support for the electrode and was allowed to dry overnight.

## Intra-cranial surgical procedures

All animals were anesthetized using isoflurane in $O_2$ (5% induction, 1% maintenance) and implanted with a chronic voltammetry microelectrode aimed at the NAc core (+1.3 AP,+1.4 ML, −6.9 DV), an ipsilateral bipolar stimulating electrode (Plastics One) in the medial forebrain bundle (−2.8 AP,+1.7 ML, −8.8 DV), and a contralateral Ag/AgCl reference electrode (Sigma-Aldrich). The reference electrode and anchoring screws were stabilized using a thin layer of dental cement (Dentsply), leaving the holes for the stimulating and recording electrodes unobstructed. The stimulating and recording electrodes were attached to a constant current isolator (A-M Systems) and voltammetric amplifier, respectively, and lowered to the most dorsal point of the target region (−6.6 DV for the working electrode and −8.5 DV for the stimulating electrode). At this depth, a triangular voltammetric input waveform (−0.4 to +1.3 V vs. Ag/AgCl, 400 V/s; *Heien et al., 2003*) was applied to the recording electrode at 60 Hz for 30 min and then reduced to 10 Hz for the remainder of the surgery. Electrical stimulation (24 biphasic pulses, 60 Hz, 120 µA) was applied to the stimulating electrode in order to evoke DA release, which was monitored at increasing depths by the recording electrode. If neither an evoked change in DA nor a physical response (whisker movement or blinking) was observed, the stimulating electrode was lowered by 0.05 mm until a response was achieved or to a maximum depth of 8.8 mm. The working electrode was then lowered by 0.05 mm until DA release was observed or to a maximum depth of 6.9 mm. Once electrically-evoked DA release was detected in the NAc core, a thin layer of dental cement was used to secure the stimulating and recording electrodes in place. A Ginder implant (Ginder Scientific; constructed in house) was connected to the reference, stimulating, and recording electrodes and fully insulated using dental cement, leaving only the screw-top connector exposed, in order to reduce noise and prevent loss of connectivity during behavioral training. Animals then received post-operative care: subcutaneous injection of 5 mL saline containing 0.04 mL carprofen (Rimadyl), topical application of lidocaine cream to the surgical area, and placement on a heating pad until full consciousness was regained. Animals were also given antibiotic treatment with Cephlexin orally twice daily post-surgery for two weeks to prevent infection of the surgical site. All subjects were allowed a month for full recovery and stabilization of the electrode before experimentation.

## Pavlovian social distress paradigm

FSCV recordings were collected in a modified shuttle box chamber (16 in x 6.25in x 8.375 in; WDH; Med Associates; n = 8 rats). A modified guillotine door with wire mesh covering the opening divided the chamber in two equal compartments. Rats could see, smell and hear each other. Each trial began with illumination of a houselight (*Figure 1A–D*). Five seconds later, one of three auditory cues (the 'outcome cue') was emitted for 5 (i.e., tone, white noise, or clicker counterbalanced across rats). One auditory cue indicated that reward would be delivered (i.e., reward trial), the second cue signaled that shock would be administered (i.e., shock trials), and the third cue (i.e. neutral) indicated that neither reward nor punishment would occur. After 5 s, the auditory cue was terminated simultaneously with the illumination of one of the two directional lights. This 'directional' cue informed the rats which side of the cage (random 50/50) would lead to a positive (reward), negative (footshock) or neutral outcome (nothing). After 5 s, the directional light extinguished, at which time, reward or punishment or nothing was administered to the side of the box that had been illuminated by the directional cue. The shock consisted of two 250 ms shocks (0.56 mA) spaced 2 s apart, starting 10 ms after offset of the directional light. Each session consisted of a total of 65 trials during which the recording rat experienced these events while 'alone' in the chamber and 65 trials during which the recording rat and paired conspecific where 'together' (the order of alone and together trial blocks were counterbalanced across sessions). This paradigm was completely Pavlovian, thus rats had no control over what outcomes would occur or which rat would receive them.

An infrared beam was placed at the entrance to the food cup on the recording rat's side of the cage. This beam was disrupted upon entry of the rat's nose into the food cup, and beam breaks served as a quantitative measure of reward seeking. In our Med Associates boxes we sampled every 10 ms to determine if the beam in the food cup was broken throughout the entire trial. For analysis, these data were aggregated as proportions across 1 s bins (i.e. divided by the number of possible breaks per second to yield a percentage). Cameras were positioned facing the recording rat. DA and video analysis focused on four trial epochs lasting five seconds in length: house light; cue tone;

directional light; and outcome. Freezing (sudden cessation of movement) and approach toward the mesh divider were assessed during these periods by two independent observers. Only sessions that contributed to the DA analysis were scored.

## Instrumental social distress task

A different set of rats (n = 9) were used to collect FSCV data during performance of the Instrumental Social Distress Task (*Figure 6A–D*). Experiments were conducted in the same modified shuttle box chambers as described above, except these boxes were equipped with a retractable lever in the recording rat's chamber, counterbalanced across pairs. On each day, rats were placed on one side of the chamber and allowed to instrumentally respond until approximately 100 pellets were earned or 45 min had elapsed. During conditioning, a cue light turned on and a lever remained extended until a response at the lever occurred. No auditory stimuli were presented. A lever press resulted in offset of the cue light, reward delivery into a magazine, and lever retraction. Conditioning sessions took place for 9 days after the rat acquired an instrumental response. For each subject, a conspecific rat was placed on the opposite side of the chamber on the last two days of conditioning. As above, rats could see, hear and smell each other through the mesh divider.

On day 10, the distress task was introduced in which a lever press resulted in lever retraction, reward delivery, and one of three outcomes: no-shock (reward only), foot shock to engaged rat (shock-self), or footshock to conspecific rat (shock-other; *Figure 6A–D*). The three trial types were signaled by discrete auditory cues (tone, white noise, clicker), which began 5 s prior to lever extension and terminated upon a lever press. A lever press for all trials resulted in immediate reward delivery and (on certain trials) an aversive footshock (3 s, 0.56 mA) 2 s after the lever press. Auditory cues were counterbalanced across animals. During trials in which the animal did not press, the lever retracted and auditory cue terminated 5 s after lever extension; reward and shock did not occur. Trials were randomly delivered in a session and the ratio of no-shock, shock-self, and conspecific foot-shock (shock-other) trials was 6:1:1 (shock trials occurring 12.5 percent of the time each, and no-shock trials occurring 75 percent of the time). The lower percentage of shock trials prevented an overall suppression of responding over all trial types (i.e., too many shock trials might cause rats to quit pressing all together). Both the recording rat and conspecific were placed in the chamber in opposing compartments separated by the previously described divider and habituated for 5 min prior to starting behavior. Daily recording sessions consisted of 100 total trials. Video recording was used to monitor behaviors of the implanted animal and conspecific.

Percent lever pressing and reaction times (lever extension to lever press) were averaged for each session and then across sessions. Video recordings were used to further assess the behavior of the recording rat and the conspecific. An approach by the recording rat to the conspecific was included in analysis if the animal's face was visible in the divider opening (camera angle down, and towards recording rat) during the time between lever retraction and light off at the end of the trial (5 s).

## Fast-scan cyclic voltammetry

For recordings, animals were connected to a head-mounted voltammetric amplifier (current-to-voltage converter) and a commutator (Crist Instruments) mounted above the recording chamber. During each session, an electrical potential was applied to the recording electrode in the same manner as described above (see *Intra-cranial surgical procedures*). In order to detect changes in dopaminergic concentration over time, the current at its peak oxidation potential was plotted for successive voltammetric scans and background signal was subtracted. Two PC-based systems, fitted with PCI multifunction data acquisition cards and software written in LabVIEW (National Instruments), were used for waveform generation, data collection, and analysis. The signal was low-pass filtered at 2,000 Hz. Event timestamps from Med Associates were recorded, in order to analyze behaviorally relevant changes in DA release.

Dopamine was identified by its stereotypical and specific cyclic voltammogram signature. Behaviorally-evoked DA signals met electrochemical criterion if the cyclic voltammogram was highly correlated to that of the DA templates produced during the training set. The training set is a template containing six each of background-subtracted cyclic voltammograms and corresponding calibrated concentrations for both DA and pH extracted from data pooled across animals acquired during electrical stimulations that are known to evoke DA release (stimulation at 1V: 30 Hz, 6 pulses; 30 Hz, 12

pulses; 30 Hz, 24 pulses; 60 Hz, 6 pulses; 60 Hz, 12 pulses; 60 Hz, 24 pulses). Voltammetric data was analyzed using software written in LabVIEW and MATLAB. A principal component regression algorithm (a MATLAB subroutine of the Tar Heel CV FSCV software) was used to extract the DA component from the raw voltammetric data (*Heien et al., 2003*; *Keithley et al., 2009*). Eigenvalues (principal components) were calculated to describe relevant components of our training set, and we performed multivariate regression analysis to determine a correlation coefficient to describe our recorded behavioral data versus the training set. The number of factors we selected to keep in our PCA analysis accounted for >99% of the variance (at least 3, but usually 4–5 factors were kept). Factor selection was an important step, as retaining more factors than we needed would have added unwanted noise to our data but retaining too few could have discarded potentially meaningful information (*Kramer, 1998*). Importantly, the exact same method was applied to each trial-type (e.g., neutral, reward, and shock) allowing for statistically sound comparisons between conditions.

We also used the residual to examine the quality of the fit. In general, the residual is the difference between the experimental observation and the predicted value derived from a model/template (our regression values) and is a measure of the unknown portion of the signal that is not accounted for by the principal components of the regression. This is important when considering the accuracy and the applicability of the model and is important for identifying possible interfering molecules or noise (such as drift). The sum of squares of the difference between the template and the experimental data is the residual value (Q) and the threshold Qa establishes whether the retained principal components provide a satisfactory description of the experimental data; the discarded principal components thus provide a measure of noise (*Heien et al., 2003*; *Keithley et al., 2009*; *Kramer, 1998*). We use this Qa measure in combination with our regression analysis to establish our concentration corrections. Chemometric analysis as applied here, is a widely-used analytical method that separates changes in current that are caused by DA release from those caused by pH shifts or other electrochemical contaminants by comparing eigenvalues derived from stimulated DA release and changes in pH to those derived from behavioral release (*Cheer et al., 2007*; *Flagel et al., 2011*; *Heien et al., 2003*; *Keithley et al., 2009*; *Kramer, 1998*; *Phillips et al., 2003*; *Wightman et al., 2007*).

## Histology

Following the completion of the study, animals were terminally anesthetized with an overdose of isoflurane (5%) and transcardially-perfused with saline and 4% paraformaldehyde. Brain tissue was removed and post-fixed with paraformaldehyde. Brains were then placed in 30% sucrose solution for 72 hr and sectioned coronally (50 μm) using a microtome. Tissue slices were mounted onto slides and stained with thionin for histological reconstruction.

## Acknowledgements

This work was supported by University of Maryland College Park/School of Medicine Seed grant and funds to MR and JFC from NIMH (MH103806, R01MH112504). Correspondence and requests for materials should be addressed to MR (email: mroesch@umd.edu).

## Additional information

### Funding

| Funder | Grant reference number | Author |
|---|---|---|
| National Institute of Mental Health | R01MH112504 | Joseph F Cheer Matthew Roesch |
| National Institute of Mental Health | MH103806 | Joseph F Cheer Matthew Roesch |

The funders had no role in study design, data collection and interpretation, or the decision to submit the work for publication.

## Author contributions

Nina T Lichtenberg, Brian Lee, Conceptualization, Data curation, Formal analysis, Supervision, Funding acquisition, Investigation, Visualization, Methodology, Writing—original draft, Project administration, Writing—review and editing; Vadim Kashtelyan, Conceptualization, Data curation, Formal analysis, Supervision, Investigation, Visualization, Methodology, Writing—original draft, Project administration, Writing—review and editing; Bharadwaja S Chappa, Henok T Girma, Elizabeth A Green, Shir Kantor, Dave A Lagowala, Matthew A Myers, Danielle Potemri, Meredith G Pecukonis, Robel T Tesfay, Michael S Walters, Data curation, Formal analysis, Investigation, Writing—review and editing; Adam C Zhao, Data curation, Formal analysis, Methodology, Writing—review and editing; R James R Blair, Joseph F Cheer, Conceptualization, Investigation, Writing—review and editing; Matthew R Roesch, Conceptualization, Resources, Data curation, Formal analysis, Supervision, Funding acquisition, Investigation, Visualization, Methodology, Writing—original draft, Project administration, Writing—review and editing

## Author ORCIDs

Matthew R Roesch (iD) http://orcid.org/0000-0003-2854-6593

## Ethics

Animal experimentation: All experiments were approved by the University of Maryland College Park Institutional Animal Care and Use Committee (R-15-34; R-JUL-18-37) under university and NIH guidelines.

## Decision letter and Author response

Decision letter https://doi.org/10.7554/eLife.38090.019
Author response https://doi.org/10.7554/eLife.38090.020

# Additional files

## Supplementary files

• Transparent reporting form
DOI: https://doi.org/10.7554/eLife.38090.015

## Data availability

Data has been uploaded to Digital Repository of the University of Maryland (https://dx.doi.org/10.13016/M2804XP75).

The following dataset was generated:

| Author(s) | Year | Dataset title | Dataset URL | Database and Identifier |
|---|---|---|---|---|
| Nina T Lichtenberg, Brian Lee, Vadim Kashtelyan, Bharadwaja S Chappa, Henok T Girma, Elizabeth A Green, Shir Kantor, Dave A Lagowala, Matthew A Myers, Danielle Potemri, Meredith G Pecukonis, Robel T Tesfay, Michael S Walters, Adam C Zhao, James R Blair, Joseph F Cheer, Matthew Roesch | 2018 | Data from Rat behavior and dopamine release are modulated by conspecific distress | https://dx.doi.org/10.13016/M2804XP75 | Digital Repository at the University of Maryland, 10.13016/M2804XP75 |

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
