## [Decision Letter]

Thank you for sending your article entitled "Rat behavior and dopamine release are modulated more by self-interest than conspecific distress" for peer review at *eLife*. Your article has been evaluated by two peer reviewers, one of whom is a member of our Board of Reviewing Editors, and the evaluation has been overseen by a Senior Editor.

Summary:

Several previous studies have presented evidence supporting prosocial or empathetic behaviors in non-human animals including rats. However, neural processes underlying these behaviors remain poorly understood. The authors measured dopamine release in the nucleus accumbens core in rats performing in two behavioral paradigms designed to study empathetic/prosocial behaviors. In a Pavlovian version of the task, cues indicated unavoidable shock or reward given to the self or a conspecific. In another task, rats can choose not to press a lever to refrain from obtaining reward to avoid a shock when a shock is expected to the self or a conspecific. The authors conclude that while rats showed some signs of empathetic behaviors, rats' behavior as well as dopamine responses primarily depended on their own outcome but not that of a conspecific.

The authors developed interesting behavioral paradigms, and addressed very interesting questions. This study provides a number of important results. However, the reviewers raised a number of substantive concerns including the generality of the results, at least as the main conclusions are presented in the current manuscript, and some issues in the data analysis. In particular, the authors emphasize the lack of empathetic behaviors and dopamine signals reflecting primarily self-interest, which can be seen as negative results. The largely negative nature of the results leaves open whether these results are due to a problem in the experimental designs or general insights. Second, some of the analyses were not convincing. For instance, some data appear to support empathetic behavior (see reviewer 2's comment #2).

The reviewers thought that the manuscript contains interesting results such as (1) the greatly diminished inhibition of dopamine release in the self-shock together condition. and (2) the decreased lever pressing that saved a conspecific from a shock. The reviewers suggest that the authors emphasize these 'positive' results more.

Essential revisions:

1) Although the main point the authors try to make is that rats are more 'self-interested' than 'empathetic', the authors fail to make the comparison that allows them to answer that question, i.e., they never compare self trials vs. conspecific trials. Instead, they limit most of their analysis to comparing the shock and reward trials relative to neutral trials.

For instance in Figure 3, which contains data from the Pavlovian assay, the authors compare the% freeze in shock trial to the neutral trial type (Figure 3A and Figure 3C) in both the self and other condition, leading the authors to conclude, "effects on freezing were absent during 'other' trials; significant increases in freezing were not observed on shock trials […] in our study, rats did not display 'empathy' defined by freezing behavior that occurs during observation of another rat freezing". Not only is this finding in direct conflict with the existing literature on empathetic behavior in rodents, it is not even indicative of what the authors observe. In fact, if the authors were to compare absolute freeze times between self-shock together and conspecific-shock together trial types, i.e., compare the closed red bar in Figure 3A with the open red bar in Figure 3C, it appears that rats indeed freeze the *same* amount to self shock and conspecific shock. In addition, the authors should use statistical methods that allow them to look at interaction of the trial conditions (self vs. conspecific) such as a multifactor ANOVA for both the behavioral and dopamine data as opposed to pairwise Wilcoxon comparisons.

Likewise, the dopamine data from the Pavlovian assay for self and conspecific trials are not even on the same figures (Figures 4 and 5 respectively).

2) While we understand that the current experimental design allows the authors to look at how rats respond to unavoidable shock to self or a nearby conspecific, the disadvantage of this design is that these trial types are anti-correlated; if the conspecific rat is shocked, then the recorded rat is not. Therefore, the recorded rat could remain in a state of uncertainty or anxiety making it impossible to disassociate 'relief' from 'empathy'. In fact, consistent with this, the authors see an increase in dopamine release when the shock was delivered to the conspecific, which is suggestive that the recorded animal might sense 'relief' that they were not being shocked. This makes the findings harder to interpret. Therefore the conclusions require a control condition in which dopamine activity is recorded where the rats observe the conspecific getting shocked but are not self-shocked.

3) The manuscript contains many potentially interesting observations. However, the authors' main conclusion – that behavior and dopamine signals better reflected the value associated with benefitting oneself as opposed to the conspecific – is largely negative in nature. This can be a general tendency of the rat (so that the authors can make conclusions that are not specific to the current experimental conditions), or this can be due to the particular behavioral conditions that the authors used (so the conclusion cannot be generalized). Because the results that the authors emphasize are largely negative in nature, it raises the issue of generality. Because of this, it appears to be difficult to draw strong conclusions from the results. Nonetheless, this study involved many experiments from which the authors make various potentially interesting observations. For instance, I found the observation of greatly diminished dopamine release inhibition in the self-shock together condition quite interesting. Furthermore, I found the decreased lever pressing that saved a conspecific from a shock very interesting. I encourage the authors to emphasize these strong conclusions first.

4) In general, it appears that the methods are lacking in details, to point out a few instances:

It is confusing what% beam breaks translates to, authors state "For analysis, these data were aggregated as proportions across 1-second bins (i.e. divided by the number of possible breaks per second to yield a percentage)." What does this value "number of possible breaks per second" correspond to?

It is not clear how freezing is quantified in the paper

- hand scored by an observer or automated pixel correlations?

- are values averaged for each animal or each session and for the entire duration of epochs?

How is% approach quantified?

- is this value handscored?

- what does% approach refer to, is it the proportion of time each trial that they spend near the mesh? Or the proportion of trials that the rats visit the mesh? kindly elucidate.

What is the duration of the shock in the two tasks, does it last the entire 5s of the outcome epoch?

The authors repeatedly use the Wilcoxon test for multiple comparisons (For e.g. Figure 3), are the p-values corrected for multiple comparisons? If the answer is yes, it should be explicitly stated. Alternatively, the better approach might be to use a multifactor ANOVA.

5) In Figure 7, the authors do not plot the data from the lever press+conspecific shock and the lever press+self shock (data preceding shock) trial types. This is important because it will be informative to see how dopamine responses to reward (sugar pellet) change with social context. For example are reward responses more subdued in the lever press + conspecific shock condition relative to lever press + no shock condition?

6) For the data in Figure 7, the authors should use an unequal variances t-test which is less prone to type 1 errors than the Wilcoxon test. This is particularly important given that the variance is much larger for the shock-self and the shock other trials relative to the no shock trials (Figure 7E).

7) There appears to be a lot of variability in the dopamine signal both across animals and across sessions (Figures 4E and 5E). Does this variability reflect instability in the recordings or does this correlate with the animal's behavior?

8) Prior studies have shown that some rats are more "empathetic" than others (For e.g. in Ben-Ami Bartal et al., 2011, some rats are openers and others non-openers). Do the authors observe this in their data? It would be interesting for the readers to see the variability in the behavioral data. It would be useful to plot the behavioral parameters by session and animal, like the authors did with the dopamine data.

9) Also, the authors should plot averaged responses (both behavior and dopamine activity) with error bars (Figures 2A, B; 4A, 5A, 7A)? Additionally, they should show trial-by-trial data in order for the reader to evaluate the consistency.

10) While authors allude to this briefly in the Discussion, the existing literature shows that familiarity plays a large role in the amount of prosocial behavior animals exhibit (rats, voles etc. are more prosocial to their cage-mates/partners relative to strangers). In contrast the animals used in this study have never been cagemates and never been in direct contact with each other. This could significantly modulate the extent or pro-social behaviors exhibited by the animal. The authors should expand on this in their Discussion.

---

## [Author Response]

Essential revisions:1) Although the main point the authors try to make is that rats are more 'self-interested' than 'empathetic', the authors fail to make the comparison that allows them to answer that question, i.e., they never compare self trials vs. conspecific trials. Instead, they limit most of their analysis to comparing the shock and reward trials relative to neutral trials.

We apologize for not making direct comparisons between self vs. conspecific trials. We now make these comparisons as in the instance described below.

For instance in Figure 3, which contains data from the Pavlovian assay, the authors compare the% freeze in shock trial to the neutral trial type (Figure 3A and Figure 3C) in both the self and other condition, leading the authors to conclude, "effects on freezing were absent during 'other' trials; significant increases in freezing were not observed on shock trials […] in our study, rats did not display 'empathy' defined by freezing behavior that occurs during observation of another rat freezing". Not only is this finding in direct conflict with the existing literature on empathetic behavior in rodents, it is not even indicative of what the authors observe. In fact, if the authors were to compare absolute freeze times between self-shock together and conspecific-shock together trial types, i.e., compare the closed red bar in Figure 3A with the open red bar in Figure 3C, it appears that rats indeed freeze the same amount to self shock and conspecific shock. In addition, the authors should use statistical methods that allow them to look at interaction of the trial conditions (self vs. conspecific) such as a multifactor ANOVA for both the behavioral and dopamine data as opposed to pairwise Wilcoxon comparisons.

Thank you for the comment. We have reported ANOVAs as suggested and report that freezing did not differ between together-self-shock and together-conspecific-shock trial types. As suggested above, we ran a 3 factor ANOVA (together vs. alone; self vs. other; trial-type (i.e., reward, neutral, shock) on freezing during the directional cue. We see significant main effects of together/alone (F(1,396) = 4.44; p < 0.05) and trial-type (F(2,396) = 16.48; p < 0.05), as well as a significant interaction between self/other and trial-type (F(2,396) = 3.3; p < 0.05). Thus, rats froze more on shock trials when alone compared to when they were together. Further, post-hoc ttests demonstrate that freezing was stronger on self-shock trials when alone compared to when rats were together (t(33) = 2.54; p < 0.05) and that freezing did not significantly differ on together-self-shock and together-other-shock trials as suggested by the reviewer (t(33) = 0.19; p = 0.85). Thank you. We have added these comparisons and statistics to the text.

Likewise, the dopamine data from the Pavlovian assay for self and conspecific trials are not even on the same figures (Figures 4 and 5 respectively).

It was difficult to put all the DA figures into one plot because there were 12 different trial-types in total. We ensured to keep the x and y axis, as well as the format and size of the plots, the same so that they could be compared to each other. Also, the analysis (i.e., distributions in B-I) in those figures examined different epochs, thus we kept them separate to increase clarity and avoid irrelevant comparisons. With that said, we now compare conspecific and self-trial types. Interestingly, as the reviewer points out, the observation that rats freeze the same level on together-shock-other as they do on together-shock-self trials is also true for DA release during the directional cue. In the 3 factor ANOVA (together vs alone; self vs other; trial-type (i.e., reward, neutral, shock), there were main effects of self/other (F(1,468) = 17.53; p < 0.05) and outcome (F(1,468) = 78.25; p < 0.05), and an interaction between self/other and outcome (F(2,468) = 5.14). Post-hoc ttests yield a significant difference between alone-shock-self and together-shock-self (t(39) = 3.53; p < 0.05) and no significant difference between together-self-shock and together-conspecific-shock (t(39) = 1.95; p = 0.06). These results have been added to the text.

2) While we understand that the current experimental design allows the authors to look at how rats respond to unavoidable shock to self or a nearby conspecific, the disadvantage of this design is that these trial types are anti-correlated; if the conspecific rat is shocked, then the recorded rat is not. Therefore, the recorded rat could remain in a state of uncertainty or anxiety making it impossible to disassociate 'relief' from 'empathy'. In fact, consistent with this, the authors see an increase in dopamine release when the shock was delivered to the conspecific, which is suggestive that the recorded animal might sense 'relief' that they were not being shocked. This makes the findings harder to interpret. Therefore the conclusions require a control condition in which dopamine activity is recorded where the rats observe the conspecific getting shocked but are not self-shocked.

We thank the reviewer for raising these interesting issues and we agree that in the Pavlovian task it is challenging to determine the psychological constructs driving behavior because the rats are not required to take action. However, it is important to note that:

i) We present both Pavlovian and the Instrumental tasks in the submitted manuscript. The instrumental task gives a direct behavioral readout of the rats’ decision (to press or not to press). In the instrumental task the rats did press significantly less and were slower to press on conspecific-shock trials demonstrating empathetic and prosocial behaviors consistent with other reports. We agree with the editors/reviewers that we should have emphasized this behavior more strongly and now do. From a neural standpoint, DA release during decisions not to shock the other rat mirrored DA release when the rat decided to avoid shock for himself. This too suggests that DA signals, like behavior, were ‘pro-social’. However, both lever pressing and DA release was less during conspecific-shock compared to self-shock, thus rats were more interested in saving themselves rather than the conspecific. In this task we think the signal is modulated by the degree of “empathy” exhibited by rats in the context of this task. Ultimately we don’t think that DA is signaling empathy, but represents prediction errors that are modulated by empathy. During shock avoidance it is known that DA release to cues reflects the value of cues that predict avoidable shock. Our data suggest that rats place less value on those cues when it is the conspecific that might potentially be shocked, consistent with degree of lever pressing that occurs on conspecific- versus self-shock trials.

ii) As the reviewers suggest, we also think in the context of the Pavlovian task, that the behavior and DA release patterns better reflect ‘relief’, as these results are consistent with the observation that rats freeze to a similar extent on conspecific shock trials when they are together compared to when they were alone. As the reviewers point out, if rats were more strongly displaying empathy, DA release would be inhibited to a stronger degree and rats would freeze more to the conspecific being shocked compared to when a shock is applied to the empty box. We actually think this a very interesting aspect of the data and can only be obtained when the trials are anti-correlated. Of note, in previous ‘empathy’ studies there was no cost to the animal when something bad happened to conspecific. Here, it appears that the ‘relief’ aspect stemming from the situation outweighed any signals that reflect the evaluation of the nature of the outcome from the perspective of the conspecific. As noted above, we do not believe that DA signals ‘relief’ or ‘empathy’ per se, but reflect prediction error correlates that are informed by social contexts. In this case, since the trial types are anti-correlated, DA increases because the rat we are recording from was not the one that got shocked, eliciting a positive prediction error. We now better explain points (i) and (ii) in the Discussion of the revision.

3) The manuscript contains many potentially interesting observations. However, the authors' main conclusion – that behavior and dopamine signals better reflected the value associated with benefitting oneself as opposed to the conspecific – is largely negative in nature. This can be a general tendency of the rat (so that the authors can make conclusions that are not specific to the current experimental conditions), or this can be due to the particular behavioral conditions that the authors used (so the conclusion cannot be generalized). Because the results that the authors emphasize are largely negative in nature, it raises the issue of generality. Because of this, it appears to be difficult to draw strong conclusions from the results. Nonetheless, this study involved many experiments from which the authors make various potentially interesting observations. For instance, I found the observation of greatly diminished dopamine release inhibition in the self-shock together condition quite interesting. Furthermore, I found the decreased lever pressing that saved a conspecific from a shock very interesting. I encourage the authors to emphasize these strong conclusions first.

Thank you. These experiments include a large body of work and we agree that there are many interesting observations from both studies. We were not expecting to find diminished inhibition in self-shock trials, but also find it quite fascinating, as well as other changes in behavior that occur during trials where there were consequences to the conspecific. We have now emphasized these conclusions more strongly in the text!

We understand the reviewer’s point of view that many of these findings appear to be “negative” results. We agree to a certain degree, but do not think they are completely “negative” in that we do see changes in behavior and DA release on social trials. It is an important question of whether behaviors and neural correlates generalize, but this is true of any neural recording paper that runs just one task. For these reasons we report data from two different studies, which could have easily made two different solid publications in their own right. Here we report in two different tasks that rats are not overly empathetic or prosocial and that DA release follows suit. With that said, we completely agree with the reviewer and now emphasize more strongly the ‘positive’ results, as well as discuss issues related to generality.

We agree with the reviewer’s/editor’s point that we should have not down played behaviors that correlate with ‘empathetic’ and ‘pro-social’ behavior to emphasize more of a self-interested interpretation of the data. That was a mistake. As suggested, we have changed the tone of the article throughout. Thank you for the suggestion.

4) In general, it appears that the methods are lacking in details, to point out a few instances:It is confusing what% beam breaks translates to, authors state "For analysis, these data were aggregated as proportions across 1-second bins (i.e. divided by the number of possible breaks per second to yield a percentage)." What does this value "number of possible breaks per second" correspond to?It is not clear how freezing is quantified in the paper- hand scored by an observer or automated pixel correlations?- are values averaged for each animal or each session and for the entire duration of epochs?How is% approach quantified?- is this value handscored?- what does% approach refer to, is it the proportion of time each trial that they spend near the mesh? Or the proportion of trials that the rats visit the mesh? kindly elucidate.What is the duration of the shock in the two tasks, does it last the entire 5s of the outcome epoch?The authors repeatedly use the Wilcoxon test for multiple comparisons (For e.g. Figure 3), are the p-values corrected for multiple comparisons? If the answer is yes, it should be explicitly stated. Alternatively, the better approach might be to use a multifactor ANOVA.

We have provided more detail throughout the revision and have performed multifactor ANOVAs as described above. In our Med Associates boxes we sampled every 10 ms to determine if the beam in the food cup was broken throughout the entire trial. DA and video analysis focused on four trial epochs lasting five seconds in length: house light; cue tone; directional light; and outcome. Freezing (sudden cessation of movement) and approach toward the mesh divider were assessed during these periods by two independent observers. Values were averaged within session and then across sessions. Only sessions that contributed to the DA analysis were scored. For the Pavlovian tasks, there were two 250 ms shocks (0.56 mA) spaced 2 s apart, starting 10 ms after offset of the directional light. For the instrumental task, a lever press for all trials resulted in immediate reward delivery and on shock trials an aversive footshock (3 s, 0.56 mA) 2 s after the lever press.

5) In Figure 7, the authors do not plot the data from the lever press+conspecific shock and the lever press+self shock (data preceding shock) trial types. This is important because it will be informative to see how dopamine responses to reward (sugar pellet) change with social context. For example are reward responses more subdued in the lever press + conspecific shock condition relative to lever press + no shock condition?

This is a very interesting question. Unfortunately, we did not show lever press + self-shock because of the noise in the signal induced from the shock. The signal is less noisy for conspecific shock trials, where we see no significant differences between lever press + conspecific shock condition relative to lever press + no shock condition (p = 0.08). We now report this in the text of the revision.

6) For the data in Figure 7, the authors should use an unequal variances t-test which is less prone to type 1 errors than the Wilcoxon test. This is particularly important given that the variance is much larger for the shock-self and the shock other trials relative to the no shock trials (Figure 7E).

We tested the variance and report unequal variances ttests to compare trial types where appropriate. Note however that the asterisks represent significant shifts from zero for each index. We ran the test to determine if the variances were different between shock-self and shock-other trials and they were not (p = 0.63). Also, the variances did not significantly differ between the 3 indices (p’s > 0.12).

7) There appears to be a lot of variability in the dopamine signal both across animals and across sessions (Figures 4E and 5E). Does this variability reflect instability in the recordings or does this correlate with the animal's behavior?

Recordings were stable. In our opinion the variability observed here is similar to what we have seen and others show in the literature. Here we go above and beyond by showing DA release for all sessions, not just in the form an error bars but in distributions that reflect the differences in strength between key comparisons for all the continuous data (i.e., DA and beam breaks). We have now explored correlations between behavior and DA release as suggested. See next point (#8) for further discussion regarding this concern.

8) Prior studies have shown that some rats are more "empathetic" than others (For e.g. in Ben-Ami Bartal et al., 2011, some rats are openers and others non-openers). Do the authors observe this in their data? It would be interesting for the readers to see the variability in the behavioral data. It would be useful to plot the behavioral parameters by session and animal, like the authors did with the dopamine data.

We have plotted the behavioral data similar to what we have done for DA in a way that allows us to explore correlations between behavior and DA release as suggested here and in comment 7. This has led to 3 new supplementary figures with four panels in each. The data show averages within rats and data from each session, color coded the same as in the DA figure in the main document.

One of our main findings is that DA release is less inhibited on self-shock trials when rats are together during the Pavlovian conspecific distress paradigm. We interpret this result as a consolation effect, whereby the threat of shock is not as aversive in the presence of the conspecific. We found that during the directional cue period rats tend to show less of suppression in the food well (Figure 2). In Figure 4—figure supplement 3 we now correlate differences in beam breaks and DA release observed on alone and together trials for sessions and rat averages, color coded as the DA figures were in the main text.

Figure 4—figure supplement 3 shows that in the majority of sessions there was reduced DA release during the direction cue epoch on shock trials relative to neutral (as in Figure 4) and there was a near significant correlation between the two (p = 0.066; r^2^ = 0.086). Likewise, for beam breaks into the food cup, there was a reduction in beam breaks on both alone and together trials, with effects being more pronounced when rats were alone (as in Figure 2). Here, we show that there was a significant correlation between the two (p < 0.05; r^2^ = 0.11). Lastly, we asked if DA and beam breaks were correlated for alone and together trials. Neither were significant (alone: p = 0.58; r^2^ = 0.008; together: p = 0.39; r^2^ = 0.02).

A second main finding is that DA release increased at the time of conspecific shock during the Pavlovian conspecific distress paradigm. As above we explored correlations between alone and together for both DA release and beam breaks, except here the analysis was performed during the outcome epoch (as in Figure 5). As in the main text, we see higher DA release on together-shock-other trials. Figure 5—figure supplement 1 illustrates the correlation between alone and together (p < 0.05; r^2^ = 0.11). During shock-other trials the suppression in bream breaks was weaker when together than when alone, but there was no correlation between the two (p = 0.95; r^2^ = 0.0001). Finally, there were no correlations between DA and beam breaks when alone (p = 0.18; r^2^ = 0.05) or together (p = 0.26; r^2^ = 0.03).

In the main text we show that during performance of the Instrumental conspecific distress paradigm that on shock-self and shock-other trials where shock was avoided (i.e., non-press trials) that DA release was present during the cue and during the period after the lever was extended but not pressed. In the main text we report that the increase in DA during the cue period was not significantly different between self-shock-avoid trials and no-shock reward-press trials. This suggests that rats similarly value avoiding the shock and obtaining reward as we have shown in a previous publication (Gentry et al., 2016). Although DA release is also present on trials when the conspecific is spared, it is at significantly lower levels (as in Figure 7). Figure 7—figure supplement 2A shows DA release on the shock-self and shock-other trials were correlated (p < 0.05; r^2^ = 0.59). This correlation was also present in the behavior (Figure 7—figure supplement 2B). To quantify the degree that the rats valued the reward relative to the avoiding the shock we subtracted percent lever pressing on shock trials from no-shock trials and subtracted the reaction times on shock trials from no-shock trials and then averaged them together, after dividing by the sum for each. This gave us one measure of the how much the recording rat valued saving themselves and the conspecific based on two behavior measures. The majority of the points fell above zero consistent with the Figure 6E and F showing that rats chose to press less often and were slower to press on shock-self and shock-other trials. Figure 7—figure supplement 2 shows that two were correlated (p < 0.05; r^2^ = 0.25). Thus, DA release and behavior reflect the value the rats place on avoiding shock. Although DA release was high when rats avoided shock, we found no significant correlation between behavior and DA release on self (p = 0.81; r^2^ = 0.003) and other (p = 0.36; r^2^ = 0.038) shock trials.

Overall, these new supplementary figures show the variability across rat and session as suggested by the reviewers. Further they examine correlations between self vs. other and alone vs. together at the level of DA release and behavior. Although we see interesting correlations between social and non-social trial types in both tasks, there were no direct correlations between behavior and DA release. This is not atypical for FSCV and single unit recording studies. It might not be surprising that DA isn’t directly correlated with gross motor output since we are only examining a single microdomain of DA release. One might not expect that the DA signal would be directly correlated with the motor act that the rat is currently engaged in but would better reflect how much value is being placed on what the rat is paying attention to at that time. Although that can be inferred by the actions of the animal, it is not always a one-to-one ratio, especially in complicated tasks. Further, DA correlates at the single neuron level and at the level of release are not time locked to or reflect the nature of movements but are triggered by cues that predict outcomes and outcome delivery itself, thus we might not expect to see correlations with movement. We could do further correlations but feel like it would be fishing at this point. The scatter plots described above show the variability in behavior as requested and examine its relationship to DA release for the main effects demonstrated in the paper. When examining these plots it is clear that there is not a clear division between pro- and anti-social rats, thus arbitrarily dividing rats does not seem feasible nor would provide more information than the correlations now presented in supplementary material contain. We think that future work looking at cagemates versus non-cagemates would be enlightening in this matter.

9) Also, the authors should plot averaged responses (both behavior and dopamine activity) with error bars (Figures 2A, B; 4A, 5A, 7A)? Additionally, they should show trial-by-trial data in order for the reader to evaluate the consistency.

We prefer not to plot error bars over line graphs unless absolutely necessary because it makes them very crowded. There are six lines per graph making it difficult to see. We agree that it is important to show variance that is why we go to great lengths to show all the variability in the distributions, scatters, and box/whisker plots showing individual sessions and averages across rats. We now also show scatter plots with sessions and rat info correlating behavior and DA release in 3 new supplementary figures.

10) While authors allude to this briefly in the Discussion, the existing literature shows that familiarity plays a large role in the amount of prosocial behavior animals exhibit (rats, voles etc. are more prosocial to their cage-mates/partners relative to strangers). In contrast the animals used in this study have never been cagemates and never been in direct contact with each other. This could significantly modulate the extent or pro-social behaviors exhibited by the animal. The authors should expand on this in their Discussion.

We expanded on this in the Discussion.